# HARNESSING THE POWER OF FEDERATED LEARNING IN FEDERATED CONTEXTUAL BANDITS

## ABSTRACT

Federated contextual bandits (FCB), a pivotal integration of federated learning (FL) and sequential decision-making, has garnered significant attention in recent years. Prior research on FCB can be understood as specific instantiations of a unified design principle articulated in this paper: "FCB = FL + CB". Here, FL enhances agents' performance by aggregating the information of other agents' local data to better contextual bandits (CB) policies. Nevertheless, it is evident that existing approaches largely employ tailored FL protocols, often deviating from the canonical FL framework. Consequently, even renowned algorithms like FedAvg remain underutilized in FCB, let alone other FL advancements. To bridge this gap between the canonical FL study and the FL component in FCB, our work introduces a novel FCB design, termed FedIGW, that incorporates inverse gap weighting as the CB algorithm. This design permits the integration of versatile FL protocols as long as they can solve a standard FL problem. With this flexible FL choice, FedIGW advances FCB research by enabling the utilization of the entire spectrum of FL innovations, encompassing canonical algorithmic designs (e.g., FedAvg and SCAFFOLD), convergence analyses, and valuable extensions (such as personalization, robustness, and privacy). We substantiate these claims through rigorous theoretical analyses and empirical evaluations.

## 1 INTRODUCTION

Federated learning (FL), initially proposed by McMahan et al. (2017); Konečný et al. (2016), has garnered significant attention for its effectiveness in enabling distributed machine learning with heterogeneous agents (Li et al., 2020a; Kairouz et al., 2021). As FL has gained popularity, numerous endeavors have sought to extend its applicability beyond the original realm of supervised learning, e.g., to unsupervised and semi-supervised learning (Zhang et al., 2020; van Berlo et al., 2020; Zhuang et al., 2022; Lubana et al., 2022). Among these directions, the exploration of federated contextual bandits (FCB) has emerged as a particularly compelling area of research, representing a pivotal fusion of FL and sequential decision-making, which has found various practical applications in cognitive radio and recommendation systems, among others.

Over the past several years, substantial progress has been achieved in the field of FCB (Wang et al., 2019; Li & Wang, 2022b; Li et al., 2022; 2023; Dai et al., 2023), particularly those involving varying function approximations (e.g., linear models, as discussed in Huang et al. (2021b); Dubey & Pentland (2020); Li & Wang (2022a); He et al. (2022); Amani et al. (2022)). Given the depth of existing research, it has become imperative to distill insights to guide future investigations. Consequently, this work first encapsulates the existing body of research under the seemingly straightforward yet overarching principle: "**FCB = FL + CB**." This principle asserts that one FCB design is functional provided that its employed FL protocol can update the parameters required by its adopted contextual bandits (CB) algorithm through the locally collected CB interaction data.

Through the lens of this "FCB = FL + CB" principle, the FL component in the previous FCB works is largely over-simplified. The FL protocol in many of these works is *one-shot* aggregation of some *compressed local data* per epoch (e.g., combining local estimates and local covariance matrices in the study of federated linear bandits). Admittedly, for some simple cases, such straightforward aggregation is sufficient. However, it limits the potential development of FCB for solving more complicated problems. In contrast, the canonical FL framework takes an optimization view of in-

corporating the local data through *multi-round* aggregation of *model parameters* (such as gradients). Recognizing this significant gap, this work aims to utilize the canonical FL framework as the FL component of FCB so as to harness the full power of FL studies in FCB.

We propose FedIGW – a pioneering design that demonstrates the ability to leverage a comprehensive array of FL advancements, encompassing canonical algorithmic approaches (like FedAvg (McMahan et al., 2017) and SCAFFOLD (Karimireddy et al., 2020)), rigorous convergence analyses, and critical appendages (such as personalization, robustness, and privacy). To the best of our knowledge, this marks the inaugural report of such a close connection between FL and FCB. The distinctive contributions of FedIGW can be succinctly summarized as follows:

• In the FCB setting with stochastic contexts and a realizable reward function, FedIGW employs the inverse gap weighting (IGW) algorithm for CB while versatile FL protocols can be incorporated, provided they can solve a standard FL problem (e.g., FedAvg and SCAFFOLD). These two parts iterate according to designed epochs: FL, drawing from previously gathered interaction data, supplies estimated reward functions for forthcoming IGW interactions. A pivotal advantage is that the flexible FL component in FedIGW provides substantial adaptability, meaning that existing and future FL protocols can be seamlessly leveraged.

• A general theoretical analysis of FedIGW is developed to demonstrate its provably efficient performance. The influence of the adopted FL protocol is captured through its optimization error, delineating the excess risk of the learned reward function. Notably, any theoretical breakthroughs in FL convergence rates can be immediately integrated into the obtained analysis and supply corresponding guarantees of FedIGW. Concretized results are further provided through demonstrations of the utilization of FedAvg and SCAFFOLD in FedIGW. Experimental results using real-world data with several different FL choices also corroborate the practicability and flexibility of FedIGW.

• Beyond its inherent generality and efficiency, FedIGW exhibits exceptional extensibility. Various appendages from FL studies can be flexibly integrated without necessitating alterations to the CB component. We explore the extension of FedIGW to personalized learning and the incorporation of privacy and robustness guarantees. Similar investigations in prior FCB works would entail substantial algorithmic modifications, while FedIGW can effortlessly leverage corresponding FL advancements to obtain these appealing attributes.

**Key related works.** Most of the previous studies on FCB are discussed in Sec. 2.2, and more comprehensively reviewed in Appendix B. We note that these FCB designs with tailored FL protocols in previous works sometimes can achieve near-optimal performance bounds in specific settings, while our proposed FedIGW is more practical and extendable. We believe these two types of designs are valuable supplements to each other. Additionally, while this work was being developed, the paper (Agarwal et al., 2023) was posted, which also proposes to have decoupled components of CB and FL in FCB. However, Agarwal et al. (2023) mainly focuses on empirical investigations, while our work offers valuable complementary contributions by conducting thorough theoretical analyses.

## 2 FEDERATED CONTEXTUAL BANDITS

This section introduces the problem of federated contextual bandits (FCB). A concise formulation is first provided. Then, the existing works are re-visited and a key principle of "FCB = FL + CB" is summarized, which reveals the major deficiency of existing works in connecting FL and FCB.

### 2.1 PROBLEM FORMULATION

**Agents.** In the FCB setting, a total of $M$ agents simultaneously participate in solving a contextual bandit (CB) problem. For generality, we consider an asynchronous system: each of the $M$ agents has a clock indicating her time step, which is denoted as $t_m = 1, 2, \cdots$ for agent $m$. For convenience, we also introduce a global time step $t$. Denote by $t_m(t)$ the agent $m$'s local time step when the global time is $t$, and $t(t_m, m)$ the global time step when the agent $m$'s local time is $t_m$.

Agent $m$ at each of her local time step $t_m = 1, 2, \cdots$ observes a context $x_{m,t_m}$, selects an action $a_{m,t_m}$ from an action set $\mathcal{A}_{m,t_m}$, and then receives the associated reward $r_{m,t_m}(a_{m,t_m})$ (possibly depends on both $x_{m,t_m}$ and $a_{m,t_m}$) as in the standard CB (Lattimore & Szepesvári, 2020). Each agent's goal is to collect as many rewards as possible given a time horizon.

Table 1: A compact summary of investigations on FCB with their adopted FL and CB components; a more comprehensive review is in Appendix B.

| Design Principle: FCB = FL + CB | | | |
|---|---|---|---|
| Reference | Setting | FL | CB |
| Globally Shared Full Model (See Section 3) | | | |
| Wang et al. (2019) | Tabular | Mean Averaging | AE |
| Wang et al. (2019); Huang et al. (2021b) | Linear | Linear Regression | AE |
| Li & Wang (2022a); He et al. (2022) | Linear | Ridge Regression | UCB |
| Li & Wang (2022b) | Gen. Linear | Distributed AGD | UCB |
| Li et al. (2022; 2023) | Kernel | Nyström Approximation | UCB |
| Dai et al. (2023) | Neural | NTK Approximation | UCB |
| FedIGW (this work) | Realizable | Flexible (e.g., FedAvg) | IGW |
| Globally Shared Partial Model (see Section 6.1) | | | |
| Li & Wang (2022a) | Linear | Alternating Minimization | UCB |
| Agarwal et al. (2020) | Realizable | FedRes.SGD | $\varepsilon$-greedy |
| FedIGW (this work) | Realizable | Flexible (e.g., LSGD-PFL) | IGW |

AE: arm elimination; Gen. Linear: generalized linear model; AGD: accelerated gradient descent

**Federation.** While many efficient single-agent (centralized) algorithms have been proposed for CB (Lattimore & Szepesvári, 2020), FCB targets building a federation among agents to perform collaborative learning such that the performance can be improved from learning independently. Especially, common interests shared among agents motivate their collaboration. Thus, FCB studies typically assume that the agents' environments are either fully (Wang et al., 2019; Huang et al., 2021b; Dubey & Pentland, 2020; He et al., 2022; Amani et al., 2022; Li et al., 2022; Li & Wang, 2022b; Dai et al., 2023) or partially (Li & Wang, 2022a; Agarwal et al., 2020) shared in the global federation.

In federated learning, the following two modes are commonly considered: (1) There exists a central server in the system, and the agents can share information with the server, which can then broadcast aggregated information back to the agents; and (2) There exists a communication graph between agents, who can share information with their neighbors in the graph. In the later discussions, we mainly consider the first scenario, i.e., collaborating through the server, which is also the main focus in FL, while both modes can be effectively encompassed in the proposed FedIGW design.

## 2.2 THE CURRENT DISCONNECTION BETWEEN FCB AND FL

The exploration of FCB traces its origins to distributed multi-armed bandits (Wang et al., 2019). Since then, FCB research has predominantly focused on enhancing performance in broader problem domains, encompassing various types of reward functions, such as linear (Wang et al., 2019; Huang et al., 2021b; Dubey & Pentland, 2020), kernelized (Li et al., 2022; 2023), generalized linear (Li & Wang, 2022b) and neural (Dai et al., 2023) (see Appendix B for a comprehensive review).

Upon a holistic review of these works, it becomes apparent that each of them focuses on a specific CB algorithm and employs a particular FL protocol to update the parameters required by CB. We thus can summarize a unified principle that "**FCB = FL + CB**": as long as two CB and FL components are *compatible* with each other, their integration results in a functional FCB design. In particular, the chosen FL protocol should possess the capability to effectively update the necessary parameterization in the employed CB algorithm. Conversely, the CB algorithm should provide appropriate datasets to facilitate the execution of the FL protocol. To be more specific, a periodically alternating design between CB and FL is commonly adopted: CB (collects one epoch of data in parallel) $\rightarrow$ FL (proceeds with CB data together and outputs CB's parameterization) $\rightarrow$ updated CB (collects another epoch of data in parallel) $\rightarrow \cdots$. A compact summary, including the components of FL and CB employed in previous FCB works, is presented in Table 1.

With this abstract principle, we can re-examine the existing works from a unified perspective to effectively guide future FCB designs. We particularly recognize that the FL components in the previous FCB works are not well investigated and even have some mismatches from canonical FL designs (McMahan et al., 2017; Konečný et al., 2016). For example, in federated linear bandits (Wang et al., 2019; Dubey & Pentland, 2020; Li & Wang, 2022a; He et al., 2022; Amani et al., 2022) and its extensions (Li et al., 2022; 2023; Li & Wang, 2022b; Dai et al., 2023), the adopted FL protocols typically involve the direct transmission and aggregation of local reward aggregates and covariance matrices, constituting a *one-shot aggregation* of *compressed local data* per epoch (albeit with subtle

variations, such as synchronous or asynchronous communications). Due to both efficiency and privacy concerns, such choices are rare (and even undesirable) in canonical FL studies, where agents typically communicate and aggregate their *model parameters* (e.g., gradients) over *multiple rounds*. Consequently, none of the existing FCB designs can seamlessly leverage the advancements in FL studies, including the renowned FedAvg algorithm (McMahan et al., 2017).

This disparity represents a significant drawback in current FCB studies, as it limits the connection between FL and FCB to merely philosophical, i.e., benefiting individual learning by collaborating through a federation, while vast FL studies cannot be leveraged to benefit FCB. Driven by this critical gap, this work aims to establish a closer relationship between FCB and FL through the introduction of a novel design, FedIGW, that is detailed in the subsequent sections. This approach provides the flexibility to integrate any FL protocol following the standard FL framework, which allows us to effectively harness the progress made in FL studies, encompassing canonical algorithmic designs, convergence analyses, and useful appendages.

## 3 FedIGW: Flexible Incorporation of FL Protocols

In this section, we present FedIGW, a novel FCB algorithm proposed in this work. Before delving into the algorithmic details, a more concrete system model with stochastic contexts and a realizable reward function is introduced. Subsequently, we outline the specifics of FedIGW, emphasizing its principal strength in seamlessly integrating canonical FL protocols.

### 3.1 System Model

Built on the formulation in Sec. 2, for each agent $m \in [M]$, denote $\mathcal{X}_m$ a context space, and $\mathcal{A}_m$ a finite set of $K_m$ actions. At each time step $t_m$ of each agent $m$, the environment samples a context $x_{m,t_m} \in \mathcal{X}_m$ and a context-dependent reward vector $r_{m,t_m} \in [0,1]^{\mathcal{A}_m}$ according to a fixed but unknown distribution $\mathcal{D}_m$. The agent $m$, as in Sec. 2, then observes the context $x_{m,t_m}$, picks an action $a_{m,t_m} \in \mathcal{A}_m$, and receives the reward $r_{m,t_m}(a_{m,t_m})$. The expected reward of playing action $a_m$ given context $x_m$ is denoted as $\mu_m(x_m, a_m) := \mathbb{E}[r_{m,t_m}(a_m)|x_{m,t_m} = x_m]$.

With no prior information about the rewards, the agents gradually learn their optimal policies, denoted by $\pi_m^*(x_m) := \arg\max_{a_m \in \mathcal{A}_m} \mu_m(x_m, a_m)$ for agent $m$ with context $x_m$. Following a standard notation (Wang et al., 2019; Huang et al., 2021b; Dubey & Pentland, 2020; Li & Wang, 2022a; He et al., 2022; Amani et al., 2022; Li & Wang, 2022b; Li et al., 2022; 2023; Dai et al., 2023), the overall regret of $M$ agents in this environment is

$$\text{Reg}(T) := \mathbb{E}\left[\sum_{m \in [M]} \sum_{t_m \in [T_m]} \left[\mu_m(x_{m,t_m}, \pi_m^*(x_{m,t_m})) - \mu_m(x_{m,t_m}, a_{m,t_m})\right]\right],$$

where $T_m = t_m(T)$ is the effective time horizon for agent $m$ given a global horizon $T$ and the expectation is taken over the randomness in contexts and rewards and the agents' algorithms. This overall regret can be interpreted as the sum of each agent $m$'s individual regret with respect to (w.r.t.) her optimal strategy $\pi_m^*$. Hence, it is ideal to be sub-linear w.r.t. the number of agents $M$, which indicates the agents' learning processes are accelerated on average due to federation.

**Realizablilty.** Despite not knowing the true expected reward functions, we consider the scenario that they are the same across agents and are within a function class $\mathcal{F}$, to which the agents have access. This assumption, rigorously stated in the following, is often referred to as the *realizability* assumption.

**Assumption 3.1** (Realizability). *There exists $f^*$ in $\mathcal{F}$ such that $f^*(x_m, a_m) = \mu_m(x_m, a_m)$ for all $m \in [M]$, $x_m \in \mathcal{X}_m$ and $a_m \in \mathcal{A}_m$.*

This assumption is a natural extension from its commonly-adopted single-agent version (Agarwal et al., 2012; Simchi-Levi & Xu, 2022; Xu & Zeevi, 2020; Sen et al., 2021) to a federated one. Note that it does not imply that the agents' environments are the same since they may face different contexts $\mathcal{X}_m$, arms $\mathcal{A}_m$, and distributions $\mathcal{D}_m^{\mathcal{X}_m}$, where $\mathcal{D}_m^{\mathcal{X}_m}$ is the marginal distribution of the joint distribution $\mathcal{D}_m$ on the context space $\mathcal{X}_m$. We study a general FCB setting only with this assumption, which incorporates many previously studied FCB scenarios as special cases. For example, the federated linear bandits (Huang et al., 2021b; Dubey & Pentland, 2020; Li & Wang, 2022a; He et al., 2022; Amani et al., 2022) are with a linear function class $\mathcal{F}$.

---

**Algorithm 1** FedIGW (Agent $m$)

---

**Input:** epoch number $l = 1$, reward function $\widehat{f}_m^l(\cdot, \cdot) = 0$, local dataset $\mathcal{S}_m^l = \emptyset$
1: **for** time step $t_m = 1, 2, \cdots$ **do**
2:      observe context $x_{m,t_m}$          ▷ *CB: IGW*
3:      compute $\widehat{a}_m^* = \arg\max_{a_m \in \mathcal{A}_m} \widehat{f}^l(a_m, x_{m,t_m})$ and action selection distribution

$$p_m^l(a_m | x_{m,t_m}) \leftarrow \begin{cases} 1 / \left( K_m + \gamma^l \left( \widehat{f}^l(\widehat{a}_m^*, x_{m,t_m}) - \widehat{f}^l(a_m, x_{m,t_m}) \right) \right) & \text{if } a_m \neq \widehat{a}_m^* \\ 1 - \sum_{a_m' \neq \widehat{a}_m^*} p_m^l(a_m' | x_{m,t_m}) & \text{if } a_m = \widehat{a}_m^* \end{cases}$$

4:      select action $a_{m,t_m} \sim p_m^l(\cdot | x_{m,t_m})$; observe reward $r_{m,t_m}(a_{m,t_m})$
5:      update the local dataset $\mathcal{S}_m^l \leftarrow \mathcal{S}_m^l \cup \{(x_{m,t_m}, a_{m,t_m}, r_{m,t_m}(a_{m,t_m}))\}$
6:      **if** $t_m = t_m(\tau^l)$ **then**          ▷ *FL*
7:          perform FL $\widehat{f}^{l+1} \leftarrow \texttt{FLroutine}(\mathcal{S}_m^l)$
8:          update dataset $\mathcal{S}_m^{l+1} \leftarrow \emptyset$; update epoch $l \leftarrow l + 1$
9:      **end if**
10: **end for**

---

### 3.2 ALGORITHM DESIGN

The FedIGW algorithm proceeds in epochs, which are separated at time slots $\tau^1, \tau^2, \cdots$ w.r.t. the global time step $t$, i.e., the $l$-th epoch starts from $t = \tau^{l-1} + 1$ and ends at $t = \tau^l$. The overall number of epochs is denoted as $l(T)$. In each epoch $l$, we describe the FL and CB components as follows, while emphasizing that the FL component is decoupled and follows the standard FL framework.

**CB: Inverse Gap Weighting (IGW).** For CB, we use inverse gap weighting (Abe & Long, 1999), which has received growing interest in the single-agent setting recently (Foster & Rakhlin, 2020; Simchi-Levi & Xu, 2022; Krishnamurthy et al., 2021; Ghosh et al., 2021) but has not been fully investigated in the federated setting. At any time step in epoch $l$, when encountering the context $x_m$, agent $m$ first identifies the optimal arm by $\widehat{a}_m^* = \arg\max_{a_m \in \mathcal{A}_m} \widehat{f}^l(x_m, a_m)$ from an estimated reward function $\widehat{f}^l$ (provided by the to-be-discussed FL component). Then, she randomly selects her action $a_m$ according to the following distribution, which is inversely proportional to each action's estimated reward gap from the identified optimal action $\widehat{a}_m^*$:

$$p_m^l(a_m | x_m) \leftarrow \begin{cases} 1 / \left( K_m + \gamma^l \left( \widehat{f}^l(\widehat{a}_m^*, x_m) - \widehat{f}^l(a_m, x_m) \right) \right) & \text{if } a_m \neq \widehat{a}_m^* \\ 1 - \sum_{a_m' \neq \widehat{a}_m^*} p_m^l(a_m' | x_m) & \text{if } a_m = \widehat{a}_m^* \end{cases},$$

where $\gamma^l$ is the learning rate in epoch $l$ that controls the exploration-exploitation tradeoff.

Besides being a valuable supplement to the currently dominating UCB-based studies in FCB, the main merit of leveraging IGW as the CB component is that it only requires an estimated reward function instead of other complicated data analytics, e.g., upper confidence bounds.

**FL: Flexible Choices.** By IGW, each agent $m$ performs local stochastic arm sampling and collects a set of data samples $\mathcal{S}_m^l := \{(x_{m,t_m}, a_{m,t_m}, r_{m,t_m} : t_m \in [t_m(\tau^{l-1}) + 1, t_m(\tau^l)])\}$ in epoch $l$. In order to enhance the performance of IGW in the subsequent epoch $l + 1$, an improved estimate $\widehat{f}^{l+1}$ based on all agents' data is desired. This objective aligns precisely with the aim of canonical FL studies, which aggregates local data for better global estimates (McMahan et al., 2017; Konečnỳ et al., 2016). Thus, the agents can target solving the following standard FL problem:

$$\min_{f \in \mathcal{F}} \widehat{\mathcal{L}}(f; \mathcal{S}_{[M]}^l) := \sum_{m \in [M]} (n_m / n) \cdot \widehat{\mathcal{L}}_m(f; \mathcal{S}_m^l), \tag{1}$$

where $n_m := |\mathcal{S}_m^l|$ is the number of samples in dataset $\mathcal{S}_m^l$, $n := \sum_{m \in [M]} n_m$ is the total number of samples, and $\widehat{\mathcal{L}}_m(f; \mathcal{S}_m^l) := (1/n_m) \cdot \sum_{i \in [n_m]} \ell_m(f(x_m^i, a_m^i); r_m^i)$ is the empirical local loss of agent $m$ with $\ell_m(\cdot; \cdot) : \mathbb{R}^2 \to \mathbb{R}$ as the loss function and $(x_m^i, a_m^i, r_m^i)$ as the $i$-th sample in $\mathcal{S}_m^l$.

As Eqn. (1) exactly follows the standard formulation of FL, the agents and the server can employ any protocol in canonical FL studies to solve this optimization, such as FedAvg (McMahan et al., 2017), SCAFFOLD (Karimireddy et al., 2020) and FedProx (Li et al., 2020a). These wildly-adopted

FL protocols typically perform iterative communications of local model parameters (e.g., gradients), instead of one-shot aggregations of compressed local data in previous FCB studies. To highlight the remarkable flexibility, we denote the adopted FL protocol as $\texttt{FLroutine}(\cdot)$. With datasets $\mathcal{S}_{[M]}^l := \{\mathcal{S}_m^l : m \in [M]\}$, the output function of this FL process, denoted as $\widehat{f}^{l+1} \leftarrow \texttt{FLroutine}(\mathcal{S}_{[M]}^l)$, is used as the estimated reward function for IGW sampling in the next epoch $l+1$.

The FedIGW algorithm for agent $m$ is summarized in Alg. 1. The key, as aforementioned, is that the component of FL in FedIGW is highly flexible as it only requires an estimated reward function for later IGW interactions. In particular, any existing or forthcoming FL protocol following the standard FL framework in Eqn. (1) can be leveraged as the $\texttt{FLroutine}(\cdot)$ in FedIGW.

## 4 THEORETICAL ANALYSIS: MODULARIZED PLUG-IN OF FL ANALYSES

In this section, we theoretically analyze the performance of the FedIGW algorithm, where the impact of the adopted FL choice is modularized as a plug-in component of its optimization error.

### 4.1 A GENERAL GUARANTEE

Denoting $E_m^l := t_m(\tau^l) - t_m(\tau^{l-1})$ as the length of epoch $l$ for agent $m$, $E_{[M]}^l := \{E_m^l : m \in [M]\}$ as the epoch length set, $\underline{c} := \min_{m \in [M], l \in [2, l(T)]} E_m^l / E_m^{l-1}$, $\overline{c} := \max_{m \in [M], l \in [2, l(T)]} E_m^l / E_m^{l-1}$ and $c := \overline{c}/\underline{c}$, the following global regret guarantee can be established.

**Theorem 4.1.** *Using a learning rate* $\gamma^l = O\left(\sqrt{\sum_{m \in [M]} E_m^{l-1} K_m / (\sum_{m \in [M]} E_m^{l-1} \mathcal{E}(E_{[M]}^{l-1}))}\right)$ *in epoch* $l$, *denoting* $\bar{K}^l := \sum_{m \in [M]} E_m^l K_m / \sum_{m \in [M]} E_m^l$, *the regret of FedIGW can be bounded as*

$$\text{Reg}(T) = O\left(\sum_{m \in [M]} E_m^1 + \sum_{l \in [2, l(T)]} c^{\frac{5}{2}} \sqrt{\bar{K}^l \mathcal{E}(E_{[M]}^{l-1})} \sum_{m \in [M]} E_m^l\right). \tag{2}$$

*Here* $\mathcal{E}(E_{[M]}^l)$ *(abbreviated from* $\mathcal{E}(\mathcal{F}; E_{[M]}^l)$*) denotes the excess risk of the output from the adopted* $\texttt{FLroutine}(\mathcal{S}_{[M]}^l)$ *using the datasets* $\mathcal{S}_{[M]}^l$, *whose formal definition is deferred to Definition C.1.*

It can be observed that in Eqn. (2), the first term bounds the regret in the first epoch. The obtained bounds for the regrets incurred within each later epoch (i.e., the term inside the sum over $l$ in the second epoch) can be interpreted as the epoch length times the expected per-step suboptimality, which then relates to the estimation quality of $\widehat{f}^l$ and thus $\mathcal{E}(E_{[M]}^{l-1})$ as $\widehat{f}^l$ is learned with the interaction data collected from epoch $l - 1$.

### 4.2 SOME CONCRETIZED DISCUSSIONS

Theorem 4.1 is notably general in the sense that a corresponding regret can be established as long as an upper bound on the excess risk $\mathcal{E}(E_{[M]}^{l-1})$ can be obtained for a certain class of reward functions and the adopted FL protocol. In the following, we provide several more concrete illustrations, and especially, a modularized framework to leverage FL convergence analyses. To ease the notation, we discuss synchronous systems with a shared number of arms in the following, i.e., $t_m = t, \forall m \in [M]$, and $K_m = K, \forall m \in [M]$, while noting similar results can be easily obtained for general systems. With this simplification, we can unify all $E_m^l$ as $E^l$ and $\bar{K}^l$ as $K$.

To initiate the concretized discussions, we start with considering a finite function class $\mathcal{F}$, i.e., $|\mathcal{F}| < \infty$, which can be extended to a function class $\mathcal{F}$ with a finite covering number of the metric space $(\mathcal{F}, l_\infty)$. In particular, the following corollary can be established via establishing $\mathcal{E}(n_{[M]}) = O(\log(|\mathcal{F}|n)/n)$ in the considered case as in Lemma D.2.

**Corollary 4.2** (A Finite Function Class). *If* $|\mathcal{F}| < \infty$ *and the adopted FL protocol provides an exact minimizer for Eqn. (1) with quadratic losses, with* $\tau^l = 2^l$, *FedIGW incurs a regret of* $\text{Reg}(T) = O(\sqrt{KMT \log(|\mathcal{F}|MT)})$ *and a total* $O(\log(T))$ *calls of the adopted FL protocol.*

We note that the obtained regret approaches the optimal regret $\Omega(\sqrt{KMT \log(|\mathcal{F}|)/\log(K)})$ of a single agent playing for $MT$ rounds (Agarwal et al., 2012) up to logarithmic factors, which demonstrates the *statistical efficiency* of the proposed FedIGW. Moreover, the total $O(\log(T))$ times call

of the FL protocol indicates that only a limited number of agents-server information-sharing are required, which further illustrates its *communication efficiency*.

As the finite function class is not often practically useful, we then focus on the canonical FL setting that each $f \in \mathcal{F}$ is parameterized by a $d$-dimensional parameter $\omega \in \mathbb{R}^d$ as $f_\omega$, e.g., a neural network. To facilitate discussions, we abbreviate $\mathcal{S} := \mathcal{S}_{[M]}$ while denoting $\omega_{\mathcal{S}}^* := \arg\min_\omega \widehat{\mathcal{L}}(f_\omega; \mathcal{S})$ as the empirical optimal parameter given a fixed dataset $\mathcal{S}$ and $\widehat{\omega}_{\mathcal{S}}$ as the output of the adopted FL protocol. We further assume $f^*$ is parameterized by the true model parameter $\omega^*$, and for a fixed $\omega$, define $\mathcal{L}(f_\omega) := \mathbb{E}_{\mathcal{S}}[\widehat{\mathcal{L}}(f_\omega; \mathcal{S})]$ as its expected loss w.r.t. the data distribution.

Following standard learning-theoretic analyses, the key task excess risk $\mathcal{E}(\mathcal{F}; n_{[M]})$ can be bounded via a combination of errors stemming from optimization and generalization.

**Lemma 4.3.** *If the loss function $l_m(\cdot; \cdot)$ is $\mu_f$-strongly convex in its first coordinate for all $m \in [M]$, it holds that $\mathcal{E}(\mathcal{F}; n_{[M]}) \leq 2\left(\varepsilon_{opt}(\mathcal{F}; n_{[M]}) + \varepsilon_{gen}(\mathcal{F}; n_{[M]})\right)/\mu_f$, where $\varepsilon_{gen}(\mathcal{F}; n_{[M]}) := \mathbb{E}_{\mathcal{S},\xi}[\mathcal{L}(f_{\widehat{\omega}_{\mathcal{S}}}) - \widehat{\mathcal{L}}(f_{\widehat{\omega}_{\mathcal{S}}}; \mathcal{S})]$ and $\varepsilon_{opt}(\mathcal{F}; n_{[M]}) := \mathbb{E}_{\mathcal{S},\xi}[\widehat{\mathcal{L}}(f_{\widehat{\omega}_{\mathcal{S}}}; \mathcal{S}) - \widehat{\mathcal{L}}(f_{\omega_{\mathcal{S}}^*}; \mathcal{S})]$.*

For the generalization error term $\varepsilon_{gen}(\mathcal{F}; n_{[M]})$, we can utilize standard results in learning theory (e.g., uniform convergence). For the sake of simplicity, we here leverage a distributional-independent upper bound on the Rademacher complexity, denoted as $\mathfrak{R}(\mathcal{F}; n_{[M]})$ (rigorously defined in Eqn. (4)), which provides that $\varepsilon_{gen}(\mathcal{F}; n_{[M]}) \leq 2\mathfrak{R}(\mathcal{F}; n_{[M]})$ using the classical uniform convergence result (see Lemma D.5). We do not further particularize this upper bound while noting it can be specified following standard procedures (Mohri et al., 2018; Bartlett et al., 2005).

On the other hand, the optimization error term $\varepsilon_{opt}(\mathcal{F}; n_{[M]})$ is exactly the standard convergence error in the analysis of FL protocols. Thus, once any theoretical breakthrough on the convergence of one FL protocol is reported, the obtained result can be immediately incorporated into our analysis framework to characterize the performance of FedIGW using that FL protocol. In particular, the following corollary is established to demonstrate the *modularized plug-in* of analyses of different FL protocols, where FedAvg (McMahan et al., 2017) and SCAFFOLD (Karimireddy et al., 2020) are adopted as further specific instances. To the best of our knowledge, this is the first time that convergence analyses of FL protocols can directly benefit the analysis of FCB designs.

**Corollary 4.4** (Modularized Plug-in of FL Analyses; A Simplified Version of Corollary D.6). *Under the condition of Lemma 4.3, the regret of FedIGW can be bounded as*

$$\mathrm{Reg}(T) = O\left(ME^1 + \sum_{l \in [2, l(T)]} \sqrt{K\left(\mathfrak{R}^{l-1} + \varepsilon_{opt}^l\right)/\mu_f} ME^l\right),$$

*where $\mathfrak{R}^l := \mathfrak{R}(\mathcal{F}; \{E^l : m \in [M]\})$ and using $\rho^l$ rounds of communications (i.e., global aggregations) and $\kappa^l$ rounds of local updates in epoch $l$, under a few other standard conditions,*

- *with **FedAvg** as the adopted* `FLroutine(·)`*, it holds that $\varepsilon_{opt}^l \leq \tilde{O}((\rho^l \kappa^l M)^{-1} + (\rho^l)^{-2})$;*
- *with **SCAFFOLD** as the adopted* `FLroutine(·)`*, it holds that $\varepsilon_{opt}^l \leq \tilde{O}((\rho^l \kappa^l M)^{-1})$.*

From this corollary, we can see that FedIGW enables a general analysis framework to seamlessly leverage theoretical advances in FL, in particular, convergence analyses. Thus, besides FedAvg and SCAFFOLD, when switching the FL component in FedIGW to FedProx (Li et al., 2020a), FedOPT (Reddi et al., 2020), and other existing or forthcoming FL designs, we can effortlessly plug in their optimization errors to obtain corresponding performance guarantees of FedIGW. This convenience highlights the theoretically intimate relationship between FedIGW and canonical FL studies.

Moreover, Corollary 4.4 can also guide how to perform the adopted FL protocol. As the generalization error is an inherent property that cannot be bypassed by better optimization results, there is no need to further proceed with the iterative FL process as long as the optimization error does not dominate the generalization error, which is reflected in a more particularized corollary in Corollary D.7.

**Remark 4.5** (A Linear Reward Function Class). As a more specified instance, we consider linear reward functions as in federated linear bandits, i.e., $f_\omega(\cdot) = \langle \omega, \phi(\cdot) \rangle$ and $f^*(\cdot) = \langle \omega^*, \phi(\cdot) \rangle$, where $\phi(\cdot) \in \mathbb{R}^d$ is a known feature mapping. In this case, the FL problem can be formulated as a standard ridge regression with $\ell_m(f_\omega(x_m, a_m); r_m) := (\langle \omega, \phi(x_m, a_m) \rangle - r_m)^2 + \lambda \|\omega\|_2^2$. With a properly chosen regularization parameter $\lambda = O(1/n)$, the generalization error can be bounded as $\varepsilon_{gen}(n_{[M]}) = \tilde{O}(d/n)$ (Hsu et al., 2012), while a same-order optimization error can be achieved

by many efficient distributed algorithms (Nesterov, 2003) with roughly $O(\sqrt{n}\log(n/d))$ rounds of communications. Then, with an exponentially growing epoch length, FedIGW can have a regret of $\tilde{O}(\sqrt{dMKT})$ with at most $\tilde{O}(\sqrt{MT})$ rounds of communications as illustrated in Appendix D.3, both of which are efficient with sublinear dependencies on the number of agents $M$ and time horizon $T$. It is worth noting that during this process, no raw or compressed data is communicated – only processed model parameters (e.g., gradients) are exchanged. This aligns with FL studies while is distinctive from previous designs for federated linear bandits (Dubey & Pentland, 2020; Li & Wang, 2022a; He et al., 2022), which often communicate covariance matrices or aggregated rewards.

## 5    EXPERIMENTAL RESULTS

In this section, we report the empirical performances of FedIGW on two real-world datasets: Bibtex (Katakis et al., 2008) and Delicious (Tsoumakas et al., 2008). For both experiments, we use 2-layered MLPs to approximate reward functions and adopt several different FL protocols in FedIGW, including FedAvg (McMahan et al., 2017), SCAFFOLD (Karimireddy et al., 2020), and FedProx (Li et al., 2020a). This is the first time, to the best of our knowledge, FedAvg is practically integrated with FCB experiments, let alone other FL protocols. Additional experimental details are discussed in Appendix G with more results provided, including error bars, performances with varying numbers of involved agents, and comparisons with FN-UCB (Dai et al., 2023).

The reported Fig. 1 compares the averaged rewards collected by FedIGW using different FL choices and $M = 10$ agents with two single-agent designs, where FALCON (Simchi-Levi & Xu, 2022) can be viewed as the single-agent version of FedIGW and AGR (Cortes, 2018) is an alternative strong single-agent CB baseline. It can be observed that on both datasets, FedIGW achieves better performance than the single-agent baselines with more rewards collected by each agent on average, which validates its effectiveness in leveraging agents' collaborations. Also, it can be observed that using the more developed SCAFFOLD and FedProx provides improved performance compared with the basic FedAvg, demonstrating FedIGW's capability of harnessing advances in FL protocols.

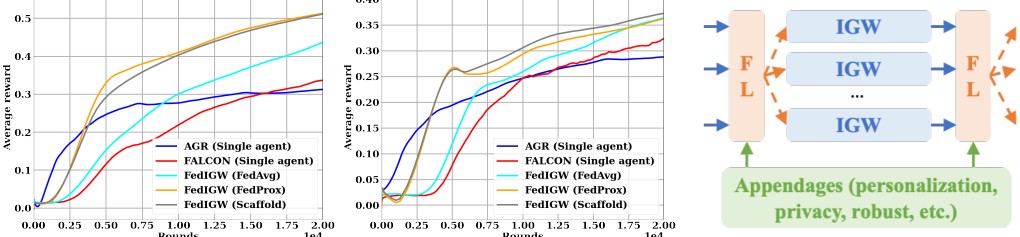

Figure 1: Experiments with Bibtex (left) and Delicious (right).        Figure 2: Flexible FL appendages

## 6    FLEXIBLE EXTENSIONS: SEAMLESS INTEGRATION OF FL APPENDAGES

Another notable advantage offered by the flexible FL choices is to bring appealing appendages from FL studies to directly benefit FCB, as illustrated in Fig. 2. In the following, we discuss how to leverage techniques of personalization, robustness, and privacy from FL in FedIGW.

### 6.1    PERSONALIZED LEARNING

In many cases, each agent's true reward function is not globally realizable as in Assumption 3.1, but instead only locally realizable in her own function class as in the following assumption.

**Assumption 6.1** (Local Realizability). *For each $m \in [M]$, there exists $f_m^*$ in $\mathcal{F}_m$ such that $f_m^*(x_m, a_m) = \mu_m(x_m, a_m)$ for all $x_m \in \mathcal{X}_m$ and $a_m \in \mathcal{A}_m$*

Following discussions in Sec. 4.2, we consider that each function $f$ in $\mathcal{F}_m$ is parameterized by a $d_m$-dimensional parameter $\omega_m \in \mathbb{R}^{d_m}$, which is denoted as $f_{\omega_m}$. Correspondingly, the true reward function $f_m^*$ is parameterized by $\omega_m^*$ and denoted as $f_{\omega_m^*}$. To still motivate the collaboration and motivated by popular personalized FL studies (Hanzely et al., 2021; Agarwal et al., 2020), we study a middle case where only partial parameters are globally shared among $\{f_{\omega_m^*} : m \in [M]\}$ while other parameters are potentially heterogeneous among agents, which can be formulated via the following assumption.

**Assumption 6.2.** *For all $m \in [M]$, the true parameter $\omega_m^*$ can be decomposed as $[\omega^{\alpha,*}, \omega_m^{\beta,*}]$ with $\omega^{\alpha,*} \in \mathbb{R}^{d^\alpha}$ and $\omega_m^{\beta,*} \in \mathbb{R}^{d_m^\beta}$, where $d^\alpha \leq \min_{m \in [M]} d_m$ and $d_m^\beta := d_m - d^\alpha$. In other words, there are $d^\alpha$-dimensional globally shared parameters among $\{\omega_m^* : m \in [M]\}$.*

A similar setting is studied in Li & Wang (2022a) for linear reward functions and in Agarwal et al. (2020) for realizable cases with a naive $\varepsilon$-greedy design for CB. For FedIGW, we can directly adopt a personalized FL protocol (such as LSGD-PFL in Hanzely et al. (2021)) to solve a standard personalized FL problem: $\min_{\omega^\alpha, \omega_{[M]}^\beta} \widehat{\mathcal{L}}(f_{\omega^\alpha, \omega_{[M]}^\beta}; \mathcal{S}_{[M]}) := \sum_{m \in [M]} n_m \widehat{\mathcal{L}}_m(f_{\omega^\alpha, \omega_m^\beta}; \mathcal{S}_m)/n$. With outputs $\widehat{\omega}^\alpha$ and $\widehat{\omega}_{[M]}^\beta$, the corresponding $M$ functions $\{f_{\widehat{\omega}^\alpha, \widehat{\omega}_m^\beta} : m \in [M]\}$ (instead of the single one $\widehat{f}$ in Sec. 3.2) can be used by the $M$ agents, separately, for their CB interactions following the IGW algorithm. Concrete results and more details can be found in Appendix E.1.

**Remark 6.3** (A Linear Reward Function Class). Similar to Remark 4.5, we also consider linear reward functions for the personalized setting with $f_m^*(\cdot) := \langle \omega_m^*, \phi(\cdot) \rangle$ and $\{\omega_m^* : m \in [M]\}$ satisfying Assumption 6.2. Then, FedIGW still can achieve a regret of $\tilde{O}(\sqrt{\tilde{d}MKT})$ with $\tilde{O}(\sqrt{MT})$ rounds of communications, where $\tilde{d} := d^\alpha + \sum_{m \in [M]} d_m^\beta$; see more details in Appendix E.1.1.

## 6.2 ROBUSTNESS, PRIVACY, AND BEYOND

Another important direction in FCB studies is to improve robustness against malicious attacks and provide privacy guarantees for local agents. A few progresses have been achieved in attaining these desirable attributes for FCB but they typically require substantial modifications to their base FCB designs, such as robustness in Demirel et al. (2022); Jadbabaie et al. (2022); Mitra et al. (2022) and privacy guarantees in Dubey & Pentland (2020); Zhou & Chowdhury (2023); Li & Song (2022).

With FedIGW, it is more convenient to achieve these attributes as suitable techniques from FL studies can be seamlessly applied. Especially, robustness and privacy protection have been extensively studied for FL in Yin et al. (2018); Pillutla et al. (2022); Fu et al. (2019) and Wei et al. (2020); Yin et al. (2021); Liu et al. (2022), respectively, among other works. As long as such FL protocols can provide an estimated function (which is the canonical goal of FL), they can be adopted in FedIGW to achieve additional robustness and privacy guarantees in FCB; see more details in Appendix E.2.

**Other Possibilities.** There have been many studies on fairness guarantees (Mohri et al., 2019; Du et al., 2021), client selections (Balakrishnan et al., 2022; Fraboni et al., 2021), and practical communication designs (Chen et al., 2021; Wei & Shen, 2022; Zheng et al., 2020) in FL among many other directions, which are all conceivably applicable in FedIGW. In addition, a recent work (Marfoq et al., 2023) studies FL with data streams, i.e., data comes sequentially instead of being static, which is a suitable design for FCB as CB essentially provides data streams. If similar ideas can be leveraged in FCB, the two components of CB and FL can truly be parallel.

## 7 CONCLUSIONS

In this work, we studied the problem of federated contextual bandits (FCB). From the perspective of the summarized principle: "FCB = FL + CB", we recognized that existing FCB designs are largely disconnected from canonical FL studies in their adopted FL protocols, which hinders the integration of crucial FL advancements. To bridge this gap, we introduced a novel design, FedIGW, capable of accommodating a wide range of FL protocols, provided they address a standard FL problem. A comprehensive theoretical performance guarantee was provided for FedIGW, highlighting its efficiency and versatility. Notably, we demonstrated the modularized incorporation of convergence analysis from FL by employing examples of the renowned FedAvg (McMahan et al., 2017) and SCAFFOLD (Karimireddy et al., 2020). Empirical validations on real-world datasets further underscored its practicality and flexibility. Moreover, we explored how advancements in FL can seamlessly bestow additional desirable attributes upon FedIGW. Specifically, we delved into the incorporation of personalization, robustness, and privacy, presenting intriguing opportunities for future research.

It would be valuable to pursue further exploration of alternative CB algorithms within FCB, e.g., Xu & Zeevi (2020); Foster et al. (2020); Wei & Luo (2021), and investigate whether the FedIGW design can be extended to more general federated RL (Dubey & Pentland, 2021; Min et al., 2023).

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

# A    ADDITIONAL DISCUSSIONS

## A.1    SOCIETAL IMPACTS

This work focuses on providing a new design for federated contextual bandits (FCB), which establishes a close relationship between FCB and FL. We do not foresee major negative societal impacts as FCB is a well-established research domain and this work largely investigates its theoretical aspects. Moreover, as discussed in Section 6.2, FedIGW can conveniently incorporate appendages from FL studies to obtain appealing properties of privacy, robustness, fairness, and beyond, which we believe can contribute to a positive societal impact.

## A.2    LIMITATIONS AND FUTURE WORKS

While this work proposes a novel, broadly applicable FCB design, i.e., FedIGW, there are still many interesting directions that are worth further exploring.

• **Paralleling CB and FL.** As mentioned in Section 2.2, the current FL studies largely focus on learning from batched and static datasets. To accommodate such protocols, FCB designs typically follow a periodically alternating scheme as shown in Fig. 3(a), which is thus the focus of this work. While such alternating designs are capable of achieving statistical and communication efficiency, there is still room for improvement: (1) the CB interactions need to wait for the completeness of a full FL process, which may be slow when computation resources are limited and communication delays are large; (2) it is desirable to use the CB data in a more timely fashion instead of accumulating to the end of an epoch.

As one variant of periodically alternating, we can have FedIGW interleave CB and FL as shown in Fig. 3(b). This approach provides some buffer to perform FL without agents waiting for its completeness. Especially, in epoch $l$, on one hand, the agents perform FL with datasets from epoch $l-1$; on the other hand, they perform CB interactions following IGW with an estimated function $\widehat{f}^{l-2}$ learned during epoch $l-1$ via datasets from epoch $l-2$. In other words, there will be one epoch delay compared with the basic form of FedIGW, while this delay is used for the FL process.

Furthermore, a better approach is to have FL and CB fully paralleled as shown in Fig. 3(c). Then, neither of them needs to wait for the other part, while CB data can be processed more timely. As mentioned in Section 6.2, we believe that the framework of FL with data streams proposed in a recent work of Marfoq et al. (2023) could be a suitable tool, as the sequential CB interactions essentially provide data steams. We believe this direction is not only worth further exploring in FCB but perhaps more importantly, calls for more investigation in FL with data streams, where FCB can also serve as an important motivation application.

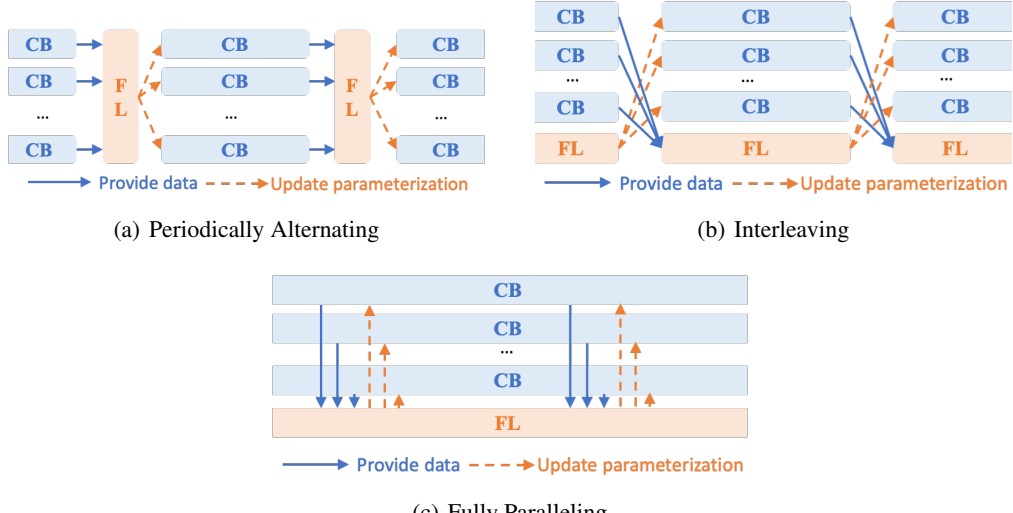

(a) Periodically Alternating            (b) Interleaving

(c) Fully Paralleling

Figure 3: Different Styles of Connecting FL and CB in FCB.

● **Incorporating other FL advances.** Given the flexible FL choice in FedIGW, although this work has provided detailed discussions on incorporating many aspects of FL advancements (including canonical algorithmic designs, convergence analysis, and useful appendages), there are still many directions worth further exploration. For example, as mentioned in Section 2.1, this work and most FCB investigations are focused on collaborating through a central server, while the case of communicating via a connected graph is less explored, where certain consensus errors commonly appear (Xin et al., 2020; Ye et al., 2020). It is worth noting that the design and analysis framework of FedIGW are both applicable in the later setting. Especially, the consensus error can be modeled as one part of the optimization error in Lemma 4.3. This further validates the value of the proposed FedIGW design and the general analysis framework while further specifications are left for future works.

Also, it would be great to leverage extra tools to save computations in the adopted FL protocol. Using local updates as in Chou et al. (2020) is one promising direction. These approaches are all feasible in FedIGW as long as the agents can obtain a learned reward function to perform IGW interactions. Their specific impacts can be captured via the established analysis framework through their own optimization errors.

● **Leveraging other CB designs.** With previous FCB studies largely focused on the CB component, this work is motivated to incorporate more advances from FL. Thus, we propose the FedIGW design which can leverage canonical protocols, convergence analyses, and flexible appendages from FL.

However, we also note that there are still many CB algorithms that remain under-explored in FCB, where UCB-based designs are dominating. For example, the simple greedy algorithm is shown to be efficient when the context generation contains certain exploration capabilities in (Han et al., 2020). Moreover, varying attempts have been made in Xu & Zeevi (2020); Foster & Rakhlin (2020); Foster et al. (2020); Zhu et al. (2022) to design generally applicable CB algorithms with tight performance guarantees, e.g., handling infinite arms. It would be interesting to investigate how to bring these designs to the federated setting and whether such connections provide new opportunities and insights.

● **Complex environments.** This work is focused on a stationary environment with stochastic rewards, which is well motivated by practical applications and commonly adopted in FCB studies. To further broaden the applicability of FCB, we believe that it is also important to study adversarial or non-stationary environments. Many advances have been made in standard single-agent bandits, e.g., Auer et al. (2002); Neu & Olkhovskaya (2020); Zierahn et al. (2023); Wei & Luo (2021), and a recent work (Yi & Vojnović, 2023) investigates the federated adversarial environment in the tabular setting. The summarized principle of "FCB = FL + CB" is conceivably still applicable, although further investigations are required to provide concrete designs and analyses.

● **Extension to RL.** It would also be meaningful to extend the current study of FCB to federated reinforcement learning (RL) as a further step in understanding the combination of FL and sequential decision-making. Some results have been reported in Dubey & Pentland (2021); Min et al. (2023); Jin et al. (2022); Fan et al. (2021); Cisneros-Velarde et al. (2023). We hope this work can serve as a starting point for more principled and generally applicable studies in federated RL.

## B   ADDITIONAL RELATED WORKS

The studies on federated multi-armed bandits (FMAB) and federated contextual bandits (FCB) can be viewed as a version of the general multi-agent bandits (Liu & Zhao, 2010; Boursier & Perchet, 2019) and parallelizing bandits (Chan et al., 2021; Karbasi et al., 2021) that is more suitable for modern applications. We provide a more detailed review in the following.

● **Tabular.** There have been many studies on cooperative designs in multi-armed bandits (i.e., the tabular setting), e.g., Hillel et al. (2013); Szorenyi et al. (2013); Landgren et al. (2016); Martínez-Rubio et al. (2019), focusing on different learning targets and different communication protocols (e.g., through a communication graph or with some randomly selected peers). Notably, in Wang et al. (2019), communication-efficient designs are proposed via periodically aggregating local estimates and performing arm elimination globally. We here also discuss another line of works on FMAB (Shi & Shen, 2021; Shi et al., 2021; Réda et al., 2022; Zhu et al., 2021; Chen et al., 2022). In their considered setting, the global rewards are (weighted) averages of local observations; however the former is not directly observable. With maximizing global rewards as the learning target, the agents

need to collaboratively perform explorations and aggregate local information. Despite the model differences, the design principle of "FCB = FL + CB" still characterizes this setting. Especially, Shi & Shen (2021); Shi et al. (2021); Réda et al. (2022); Zhu et al. (2021); Chen et al. (2022) all commonly employ UCB-based exploration designs while the adopted FL protocol is to average local sample means as global ones and to construct global confidence bounds.

• **Linear.** The most commonly studied FCB setting is federated linear bandits. There have been many investigations in this direction. Especially, different environments have been tackled in different works, e.g., the finite-armed fixed-context setting (Wang et al., 2019; Huang et al., 2021b), the finite-armed stochastic-context setting (Amani et al., 2022), the infinite-armed fixed-context setting (Salgia & Zhao, 2022), and the infinite-armed adversarial-context setting (Wang et al., 2019; Dubey & Pentland, 2020; Li & Wang, 2022a; He et al., 2022). Furthermore, many other settings, e.g., unobserved context (Lin & Moothedath, 2023), and additional properties, e.g., privacy (Dubey & Pentland, 2020; Zhou & Chowdhury, 2023), robustness (Jadbabaie et al., 2022), have been investigated. As summarized in the main paper, these works mainly select arm elimination (AE) (Lattimore & Szepesvári, 2020) or LinUCB (Abbasi-Yadkori et al., 2011) as their CB designs, which require both model estimates and confidence bounds. Thus, in their designed FL protocols, compressed local data (e.g., aggregated local rewards and covariance matrices) are often directly shared to solve a global ridge regression and to construct tighter confidence bounds. Compared with these studies, FedIGW can effectively solve the finite-armed stochastic-context setting without sharing any raw or compressed local data but only communicate processed model parameters (e.g., gradients). More detailed discussions and concrete results are provided in Appendix D.3.

A detailed comparison of the obtained regrets and the amounts of communicated real numbers is provided in Table 2. It can be observed that adapting FedIGW to the specific case of linear bandits does not provide the same near-optimal performance as in previous works. This is not a surprise as during our pursuit of generality and extendability, intuitively, certain theoretical optimalities may be inevitably sacrificed compared with previous tailored designs. Especially, the previous works are specifically optimized for the linear class of reward functions while FedIGW can handle any realizable class of realizable reward functions, which is much more general.

Table 2: A comparison of settings and results of federated linear bandits; note that FedIGW is not specifically designed and optimized to handle linear reward functions as previous designs.

| Reference | Arms | Context | Regret | # of Numbers Communicated |
|---|---|---|---|---|
| Wang et al. (2019) | Infinite | Fixed | $\tilde{O}(d\sqrt{MT})$ | $O((dM + d\log\log(d))\log(T))$ |
| He et al. (2022) | Infinite | Adversarial | $\tilde{O}(d\sqrt{MT})$ | $O(d^3M^2\log(MT))$ |
| Huang et al. (2021b) | Finite | Fixed | $\tilde{O}(\sqrt{dMT})$ | $O(d^2 + dK)M\log(T))$ |
| Amani et al. (2022)† | Finite | Stochastic | $\tilde{O}(\sqrt{dMT})$ | $O(dM\log\log(MT))$ |
| FedIGW‡ | Finite | Stochastic | $\tilde{O}(\sqrt{dKMT})$ | $O(d^2M\log(T))$ |
| FedIGW♭ | Finite | Stochastic | $\tilde{O}(\sqrt{dKMT})$ | $O(d\log(d)\sqrt{M^3T})$ |

†: assuming a homogeneous and known context distribution for all agents;
‡: solving the global ridge regression via directly sharing aggregated local rewards and covariance matrices as in the other listed works;
♭: solving the global ridge regression via distributed accelerated gradient descent;

• **Generalized Linear and Kernelized.** As extensions of the linear reward functions, Li & Wang (2022b) considers the generalized-linear class, and Li et al. (2022; 2023) study the kernelized one. The adopted basic techniques are similar to the aforementioned ones in federated linear bandits, while efforts are focused on fine-tuning communications (e.g., via Nyström approximation (Li et al., 2022; 2023)). It is worth noting that Li & Wang (2022b) invokes the distributed accelerated gradient descent algorithm to solve their considered distributed optimization with a generalized linear function class, which can be viewed as a preliminary attempt of involving FL or distributed optimization designs in FCB. However, the motivation there is the lack of a closed-form solution as in the linear case, while Li & Wang (2022b) additionally needs to share the local covariance matrices to construct better confidence bounds. This work, instead, formally proposed FedIGW which can rely only on canonical FL framework and accommodate flexible FL choices.

• **Neural.** A recent work of Dai et al. (2023) extends the advances on single-agent neural bandits (Zhou et al., 2020) to the federated setting, where the neural tangent kernel (NTK) analyses are incorporated. With NTK to "linearize" the considered over-parameterized neural network, Dai et al.

(2023) still largely follows the designs in the aforementioned federated linear bandits while some additional attempts have been made, e.g., an extra one-round averaging of model parameters besides aggregating NTK. This work, instead, takes a step further to fully leverage FL protocols, which often perform multiple (instead of one) rounds of model aggregations that are often necessary to guarantee convergence. Also, the optimization and generalization errors of a FedAvg variant with overparameterized neural networks are provided in Huang et al. (2021a), which is conceivably compatible with FedIGW for the corresponding analyses. Moreover, as shown by the additional experimental results in Appendix G.4, FedIGW empirically outperforms FN-UCB (Dai et al., 2023) on different tasks and is more computationally efficient.

## C  PROOFS FOR SECTION 4.1

### C.1  NOTATIONS

We first introduce notations that are repeatedly used in the proofs. For the output function from the adopted FL protocol, we characterize its performance via the following definition of its excess risk, which is commonly adopted in the analysis of IGW-type CB algorithms (Simchi-Levi & Xu, 2022; Sen et al., 2021; Ghosh et al., 2021).

**Definition C.1.** *Let $p_{[M]} := \{p_m : m \in [M]\}$ be a set of $M$ arbitrary independent arm selection distributions. Given an overall dataset $\mathcal{S}_{[M]} := \{\mathcal{S}_m : m \in [M]\}$ where each dataset $\mathcal{S}_m$ consists of $n_m$ training samples of the form $(x_m, a_m; r_m(a_m))$ independently and identically drawn according to $(x_m, r_m) \sim \mathcal{D}_m$, $a_m \sim p_m(\cdot|x_m)$, the federated protocol $\mathtt{FLroutine}(\mathcal{S}_{[M]}) = \{\mathtt{FLroutine}_m(\mathcal{S}_m) : m \in [M]\}$ returns a predictor $\widehat{f}(\cdot)$, and its excess risk is defined as*

$$\mathcal{E}(\mathcal{F}; n_{[M]}) := \mathbb{E}_{S_{[M]}, \xi} \left[ \sum_{m \in [M]} \frac{n_m}{n} \cdot \mathbb{E}_{x_m \sim \mathcal{D}_m^{\mathcal{X}_m}, a_m \sim p_m(\cdot|x_m)} \left[ \left( \widehat{f}(x_m, a_m) - f^*(x_m, a_m) \right)^2 \right] \right],$$

*where $n_{[M]} := \{n_m : m \in [M]\}$ and $\xi$ denotes the random source in the potentially stochastic FL algorithm. We often abbreviate $\mathcal{E}(\mathcal{F}; n_{[M]})$ as $\mathcal{E}(n_{[M]})$ to simplify notations.*

This definition measures in expectation (w.r.t. the random data generation and the stochastic FL process) how far the output of the adopted FL protocol is from the true reward function on the weighted data distribution of all agents. Note that the excess risk bound $\mathcal{E}(n_{[M]})$ would typically rely on some other parameters in the adopted FL protocol (e.g., the step size and the number of iterations in gradient-based approaches), which are currently not specified for generality.

Then, let $\Upsilon^l$ denote the sigma-algebra generated by the history up to epoch $l$, i.e., $\{(x_{m,t_m}, a_{m,t_m}, r_{m,t_m}) : m \in [M], t_m \in [t_m(\tau^l)]\}$, and the randomness in the adopted FL protocol up to epoch $l$, i.e., $\{\xi_i : i \in [l]\}$, where $\xi_i$ denotes the random source in epoch $i$. Then, we denote $l_m(t_m) := \min\{l \in \mathbb{N} : t_m \leq t_m(\tau^l)\}$ as the epoch that agent $m$'s $t_m$ belongs to. Also, let $\Psi_m := \mathcal{A}_m^{\mathcal{X}_m}$ denote the set of deterministic functions from $\mathcal{X}_m$ to $\mathcal{A}_m$ for agent $m$ and $\Psi_{[M]} := \times_{m \in [M]} \Psi_m$ the Cartesian product of $\{\Psi_m : m \in [M]\}$. Furthermore, for any action selection kernel $p_{[M]} = \{p_m : m \in [M]\}$, where $p_m(a_m|x_m)$ is the probability of selecting action $a_m \in \mathcal{A}$ given context $x_m$, and any policy $\pi_{[M]} = \{\pi_m : m \in [M]\} \in \Psi$, we define

$$V_m(p_m, \pi_m) := \mathbb{E}_{x_m \sim \mathcal{D}_m^{\mathcal{X}_m}} \left[ \frac{1}{p_m(\pi_m(x_m)|x_m)} \right],$$

$$\mathcal{R}_m(\pi_m) := \mathbb{E}_{x_m \sim \mathcal{D}_m^{\mathcal{X}_m}} \left[ f^*(x_m, \pi_m(x_m)) \right],$$

$$\widehat{\mathcal{R}}_m^l(\pi_m \mid \Upsilon^{l-1}) := \mathbb{E}_{x_m \sim \mathcal{D}_m^{\mathcal{X}_m}} \left[ \widehat{f}^l(x_m, \pi_m(x_m)) \mid \Upsilon^{l-1} \right],$$

$$\mathrm{Reg}_m(\pi_m) := \mathcal{R}_m(\pi_m^*) - \mathcal{R}_m(\pi_m),$$

$$\widehat{\mathrm{Reg}}_m^l(\pi_m \mid \Upsilon^{l-1}) := \widehat{\mathcal{R}}_{m,t_m}^l(\widehat{\pi}_m^l \mid \Upsilon^{l-1}) - \widehat{\mathcal{R}}_{m,t_m}^l(\pi_m \mid \Upsilon^{l-1}).$$

where $\widehat{\pi}_m^l(x_m) := \arg\max_{a_m \in \mathcal{A}_m} \widehat{f}^l(x_m, a_m)$ for a given $\widehat{f}^l$ (determined by $\Upsilon^{l-1}$).

The following proofs are largely inspired by the single-agent contextual bandits work (Simchi-Levi & Xu, 2022), while major changes have been made to accommodate the more complex federated system considered in this work.

## C.2 PROOFS OF THEOREM 4.1

First, the following lemma characterizes the relation between the excess errors and the selected learning rates.

**Lemma C.2.** *For all $l > 1$, it holds that*

$$\mathbb{E}_{\Upsilon^{l-1}} \left[ \sum_{m \in [M]} \frac{E_m^{l-1}}{\sum_{m' \in [M]} E_{m'}^{l-1}} \cdot \mathbb{E}_{x_m \sim \mathcal{D}_m^{\mathcal{X}_m}, a_m \sim p_m^{l-1}(\cdot | x_m)} \left[ \left( \widehat{f}^l(x_m, a_m) - f^*(x_m, a_m) \right)^2 \mid \Upsilon^{l-1} \right] \right]$$

$$\leq \mathcal{E}(\mathcal{F}; E_{[M]}^{l-1}) = \frac{\sum_{m \in [M]} E_m^{l-1} K_m}{\sum_{m \in [M]} E_m^{l-1} (\gamma^l)^2}.$$

*Proof.* The first inequality is from the Assumption C.1, while the second is based on the choice of $\gamma^l$ in Theorem 4.1, i.e.,

$$\gamma^l = \sqrt{\frac{\sum_{m \in [M]} E_m^{l-1} K_m}{\sum_{m \in [M]} E_m^{l-1} \mathcal{E}(\mathcal{F}; E_{[M]}^{l-1})}},$$

which leads to the lemma. □

Then, the following lemma bounds the estimated rewards $\widehat{\mathcal{R}}_m^l$ and true rewards $\mathcal{R}_m$.

**Lemma C.3.** *For any epoch $l > 1$, for any $\pi_m \in \Psi_m$, conditioned on $\Upsilon^{l-1}$, it holds that*

$$\left| \widehat{\mathcal{R}}_m^l(\pi_m \mid \Upsilon^{l-1}) - \mathcal{R}_m(\pi_m) \right| \leq \sqrt{V_m(p_m^{l-1}, \pi_m \mid \Upsilon^{l-1})} \sqrt{\mathcal{E}_m^{l-1}(\Upsilon^{l-1})},$$

*where $\mathcal{E}_m^{l-1}(\Upsilon^{l-1}) := \mathbb{E}_{x_m \sim \mathcal{D}_m^{\mathcal{X}_m}, a_m^{l-1} \sim p_m^{l-1}(\cdot | x_m)} \left[ \left( \widehat{f}^l(x_m, a_m^{l-1}) - f^*(x_m, a_m^{l-1}) \right)^2 \mid \Upsilon^{l-1} \right].$*

*Proof.* For simplicity, we abbreviate $\mathbb{E}_{x_m \sim \mathcal{D}_m^{\mathcal{X}_m}, a_m^{l-1} \sim p_m^{l-1}(\cdot | x_m)}[\cdot]$ as $\mathbb{E}_{x_m, a_m^{l-1}}[\cdot]$, and for any policy $\pi_m \in \Psi_m$, and any epoch $l > 1$, we define

$$\Delta_m^l(\pi_m(x_m)) := \widehat{f}^l(x_m, \pi_m(x_m)) - f^*(x_m, \pi_m(x_m))$$

which indicates that

$$\widehat{\mathcal{R}}_m^l(\pi_m \mid \Upsilon^{l-1}) - \mathcal{R}_m(\pi_m) = \mathbb{E}_{x_m} \left[ \Delta_m^l(\pi_m(x_m) \mid \Upsilon^{l-1} \right],$$

and

$$\mathbb{E}_{x_m, a_m^{l-1}} \left[ \left( \Delta_m^l(a_m^{l-1}) \right)^2 \mid \Upsilon^{l-1} \right] \geq \mathbb{E}_{x_m} \left[ p_m^{l-1}(\pi_m(x_m) | x_m) \left( \Delta_m^l(\pi_m(x_m)) \right)^2 \mid \Upsilon^{l-1} \right].$$

Furthermore, conditioned on $\Upsilon^{l-1}$, we can obtain that

$$V_m(p_m^{l-1}, \pi_m \mid \Upsilon^{l-1}) \cdot \mathbb{E}_{x_m, a_m^{l-1}} \left[ \left( \Delta_m^l(a_m^{l-1}) \right)^2 \mid \Upsilon^{l-1} \right]$$

$$= \mathbb{E}_{x_m} \left[ \frac{1}{p_m^{l-1}(\pi_m(x_m) | x_m)} \mid \Upsilon^{l-1} \right] \mathbb{E}_{x_m, a_m^{l-1}} \left[ \left( \Delta_m^l(a_m^{l-1}) \right)^2 \mid \Upsilon^{l-1} \right]$$

$$\geq \left( \mathbb{E}_{x_m} \left[ \sqrt{\frac{1}{p_m^{l-1}(\pi_m(x_m) | x_m)} \mathbb{E}_{a_m^{l-1}} \left[ \left( \Delta_m^l(a_m^{l-1}) \right)^2 \right]} \mid \Upsilon^{l-1} \right] \right)^2$$

$$\geq \left( \mathbb{E}_{x_m} \left[ \sqrt{\frac{1}{p_m^{l-1}(\pi_m(x_m) | x_m)} p_m^{l-1}(\pi_m(x_m) | x_m) \left( \Delta_m^l(\pi_m(x_m)) \right)^2} \mid \Upsilon^{l-1} \right] \right)^2$$

$$= \left( \mathbb{E}_{x_m} \left[ \left| \Delta_m^l(\pi_m(x_m)) \right| \mid \Upsilon^{l-1} \right] \right)^2$$

$$\geq \left| \widehat{\mathcal{R}}_m^l(\pi_m \mid \Upsilon^{l-1}) - \mathcal{R}_m(\pi_m) \right|^2.$$

As a result, it holds that

$$\left|\widehat{\mathcal{R}}_m^l(\pi_m \mid \Upsilon^{l-1}) - \mathcal{R}_m(\pi_m)\right| \leq \sqrt{V_m(p_m^{l-1}, \pi_m \mid \Upsilon^{l-1})}\sqrt{\mathcal{E}_m^{l-1}(\Upsilon^{l-1})},$$

where the last step we use the definition that

$$\mathcal{E}_m^{l-1}(\Upsilon^{l-1}) = \mathbb{E}_{x_m, a_m^{l-1}}\left[\left(\widehat{f}^l(x_m, a_m^{l-1}) - f^*(x_m, a_m^{l-1})\right)^2 \mid \Upsilon^{l-1}\right].$$

This concludes the proof. $\qquad\square$

Furthermore, the following lemma provides a characterization of the relation between the virtual loss $\widehat{\mathrm{Reg}}_m^l$ and the true loss $\mathrm{Reg}_m^l$.

**Lemma C.4.** *For any epochs $l \geq 1$, for any policies $\pi_{[M]} \in \Psi_{[M]}$, it holds that*

$$\sum_{m \in [M]} E_m^l \mathrm{Reg}_m(\pi_m) \leq 2 \sum_{m \in [M]} E_m^l \mathbb{E}_{\Upsilon^{l-1}}\left[\widehat{\mathrm{Reg}}_m^l(\pi_m \mid \Upsilon^{l-1})\right] + \eta^l,$$

$$\sum_{m \in [M]} E_m^l \mathbb{E}_{\Upsilon^{l-1}}\left[\widehat{\mathrm{Reg}}_m^l(\pi_m \mid \Upsilon^{l-1})\right] \leq 2 \sum_{m \in [M]} E_m^l \mathrm{Reg}_m(\pi_m) + \eta^l,$$

*with*

$$\eta^l := \frac{9c^2}{\gamma^l} \sum_{m \in [M]} E_m^l K_m.$$

*Proof.* First, we note that for $l = 1$, it holds that

$$\sum_{m \in [M]} E_m^1 \mathrm{Reg}_m(\pi_m) \leq \sum_{m \in [M]} E_m^1 \leq \eta^1 = 9c^2 \sum_{m \in [M]} E_m^1 K_m;$$

$$\sum_{m \in [M]} E_m^1 \widehat{\mathrm{Reg}}_m^l(\pi_m) = 0 \leq \eta^1 = 9c^2 \sum_{m \in [M]} E_m^1 K_m,$$

which means the lemma holds for the first epoch.

We then perform an inductive proof and start by assuming that for epoch $l-1$ and any policies $\pi_m \in \Psi_m$, it holds that

$$\sum_{m \in [M]} E_m^{l-1} \mathrm{Reg}_m(\pi_m) \leq 2 \sum_{m \in [M]} E_m^{l-1} \mathbb{E}_{\Upsilon^{l-2}}\left[\widehat{\mathrm{Reg}}_m^{l-1}(\pi_m \mid \Upsilon^{l-2})\right] + \eta^{l-1}$$

$$\sum_{m \in [M]} E_m^{l-1} \mathbb{E}_{\Upsilon^{l-2}}\left[\widehat{\mathrm{Reg}}_m^{l-1}(\pi_m \mid \Upsilon^{l-2})\right] \leq 2 \sum_{m \in [M]} E_m^{l-1} \mathrm{Reg}_m(\pi_m) + \eta^{l-1}.$$

Then, it can be observed that

$$\mathrm{Reg}_m(\pi_m) - \widehat{\mathrm{Reg}}_m^l(\pi_m \mid \Upsilon^{l-1})$$

$$= \mathcal{R}_m(\pi_m^*) - \mathcal{R}_m(\pi_m) - \left(\widehat{\mathcal{R}}_m^l(\widehat{\pi}_m^l \mid \Upsilon^{l-1}) - \widehat{\mathcal{R}}_m^l(\pi_m \mid \Upsilon^{l-1})\right)$$

$$\leq \mathcal{R}_m(\pi_m^*) - \mathcal{R}_m(\pi_m) - \left(\widehat{\mathcal{R}}_m^l(\pi_m^* \mid \Upsilon^{l-1}) - \widehat{\mathcal{R}}_m^l(\pi_m \mid \Upsilon^{l-1})\right)$$

$$= \mathcal{R}_m(\pi_m^*) - \widehat{\mathcal{R}}_m^l(\pi_m^* \mid \Upsilon^{l-1}) + \widehat{\mathcal{R}}_m^l(\pi_m \mid \Upsilon^{l-1}) - \mathcal{R}_m(\pi_m)$$

$$\overset{(a)}{\leq} \sqrt{V_m(p_m^{l-1}, \pi_m^* \mid \Upsilon^{l-1})}\sqrt{\mathcal{E}_m^{l-1}(\Upsilon^{l-1})} + \sqrt{V_m(p_m^{l-1}, \pi_m \mid \Upsilon^{l-1})}\sqrt{\mathcal{E}_m^{l-1}(\Upsilon^{l-1})}$$

$$\leq \frac{V_m(p_m^{l-1}, \pi_m^* \mid \Upsilon^{l-1})}{8c\gamma^l} + \frac{V_m(p_m^{l-1}, \pi_m \mid \Upsilon^{l-1})}{8c\gamma^l} + 4c\gamma^l \mathcal{E}_m^{l-1}(\Upsilon^{l-1})$$

$$\overset{(b)}{\leq} \frac{K_m + \gamma^{l-1}\widehat{\mathrm{Reg}}_m^{l-1}(\pi_m^* \mid \Upsilon^{l-1})}{8c\gamma^l} + \frac{K_m + \gamma^{l-1}\widehat{\mathrm{Reg}}_m^{l-1}(\pi_m \mid \Upsilon^{l-1})}{8c\gamma^l} + 4c\gamma^l \mathcal{E}_m^{l-1}(\Upsilon^{l-1}),$$

where inequality (a) is from Lemma C.3 and inequality (b) is from Lemma C.10.

Then, summing over all $M$ agents, we can obtain that

$$
\mathbb{E}_{\Upsilon^{l-1}} \left[ \sum_{m \in [M]} E_m^l \left( \text{Reg}_m(\pi_m) - \widehat{\text{Reg}}_m^l(\pi_m \mid \Upsilon^{l-1}) \right) \right]
$$

$$
\leq \frac{\sum_{m \in [M]} E_m^l K_m}{4c\gamma^l} + \frac{\gamma^{l-1}}{8c\gamma^l} \sum_{m \in [M]} E_m^l \mathbb{E}_{\Upsilon^{l-1}} \left[ \widehat{\text{Reg}}_m^{l-1}(\pi_m^* \mid \Upsilon^{l-1}) \right]
$$

$$
+ \frac{\gamma^{l-1}}{8c\gamma^l} \sum_{m \in [M]} E_m^l \mathbb{E}_{\Upsilon^{l-1}} \left[ \widehat{\text{Reg}}_m^{l-1}(\pi_m \mid \Upsilon^{l-1}) \right] + 4c\gamma^l \sum_{m \in [M]} E_m^l \mathbb{E}_{\Upsilon^{l-1}} \left[ \mathcal{E}_m^{l-1}(\Upsilon^{l-1}) \right]
$$

$$
\overset{(d)}{\leq} \frac{\sum_{m \in [M]} E_m^l K_m}{4c\gamma^l} + \frac{\overline{c}\gamma^{l-1}}{8c\gamma^l} \sum_{m \in [M]} E_m^{l-1} \mathbb{E}_{\Upsilon^{l-1}} \left[ \widehat{\text{Reg}}_m^{l-1}(\pi_m^* \mid \Upsilon^{l-1}) \right]
$$

$$
+ \frac{\overline{c}\gamma^{l-1}}{8c\gamma^l} \sum_{m \in [M]} E_m^{l-1} \mathbb{E}_{\Upsilon^{l-1}} \left[ \widehat{\text{Reg}}_m^{l-1}(\pi_m \mid \Upsilon^{l-1}) \right] + 4c\gamma^l \sum_{m \in [M]} E_m^l \mathbb{E}_{\Upsilon^{l-1}} \left[ \mathcal{E}_m^{l-1}(\Upsilon^{l-1}) \right]
$$

$$
\overset{(e)}{\leq} \frac{\sum_{m \in [M]} E_m^l K_m}{4c\gamma^l} + \frac{\overline{c}\gamma^{l-1}}{4c\gamma^l} \sum_{m \in [M]} E_m^{l-1} \text{Reg}_m(\pi_m) + \frac{\overline{c}\gamma^{l-1}}{4c\gamma^l} \cdot \eta^{l-1}
$$

$$
+ 4c\gamma^l \sum_{m \in [M]} E_m^l \mathbb{E}_{\Upsilon^{l-1}} \left[ \mathcal{E}_m^{l-1}(\Upsilon^{l-1}) \right]
$$

$$
\overset{(f)}{\leq} \frac{\sum_{m \in [M]} E_m^l K_m}{4c\gamma^l} + \frac{1}{4} \sum_{m \in [M]} E_m^l \text{Reg}_m(\pi_m) + \frac{9c^2 \sum_{m \in [M]} E_m^l K_m}{4\gamma^l} + \frac{4c^2 \sum_{m \in [M]} E_m^l K_m}{\gamma^l},
$$

where inequality (d) is from the definition $\overline{c} := \max_{m \in [M], l \in [2, l(T)]} E_m^l / E_m^{l-1}$. Inequality (e) is from the induction assumption that

$$
\sum_{m \in [M]} E_m^{l-1} \mathbb{E}_{\Upsilon^{l-1}} \left[ \widehat{\text{Reg}}_m^{l-1}(\pi_m^* \mid \Upsilon^{l-1}) \right] = \sum_{m \in [M]} E_m^{l-1} \mathbb{E}_{\Upsilon^{l-2}} \left[ \widehat{\text{Reg}}_m^{l-1}(\pi_m^* \mid \Upsilon^{l-2}) \right]
$$

$$
\leq 2 \sum_{m \in [M]} E_m^{l-1} \text{Reg}_m(\pi_m^*) + \eta^{l-1} = \eta^{l-1},
$$

$$
\sum_{m \in [M]} E_m^{l-1} \mathbb{E}_{\Upsilon^{l-1}} \left[ \widehat{\text{Reg}}_m^{l-1}(\pi_m \mid \Upsilon^{l-1}) \right] = \sum_{m \in [M]} E_m^{l-1} \mathbb{E}_{\Upsilon^{l-2}} \left[ \widehat{\text{Reg}}_m^{l-1}(\pi_m \mid \Upsilon^{l-2}) \right]
$$

$$
\leq 2 \sum_{m \in [M]} E_m^{l-1} \text{Reg}_m(\pi_m) + \eta^{l-1}.
$$

Inequality (f) is based on the definition $\underline{c} := \min_{m \in [M], l \in [2, l(T)]} E_m^l / E_m^{l-1}$, $c := \overline{c}/\underline{c}$ and $\eta^l := 9c^2 \sum_{m \in [M]} E_m^l K_m / \gamma^l$, also the assumption that $\gamma^l \geq \gamma^{l-1}$ and Lemma C.2, which indicates that

$$
\mathbb{E}_{\Upsilon^{l-1}} \left[ \sum_{m \in [M]} E_m^{l-1} \mathcal{E}_m^{l-1}(\Upsilon^{l-1}) \right] \leq \frac{\sum_{m \in [M]} E_m^{l-1} K_m}{(\gamma^l)^2}.
$$

Thus, we can obtain that

$$
\frac{3}{4} \sum_{m \in [M]} E_m^l \text{Reg}_m(\pi_m) \leq \sum_{m \in [M]} E_m^l \mathbb{E}_{\Upsilon^{l-1}} \left[ \widehat{\text{Reg}}_m^l(\pi_m \mid \Upsilon^{l-1}) \right] + \frac{\sum_{m \in [M]} E_m^l K_m}{4c\gamma^l}
$$

$$
+ \frac{25c^2 \sum_{m \in [M]} E_m^l K_m}{4\gamma^l}
$$

$$
\Rightarrow \sum_{m \in [M]} E_m^l \text{Reg}_m(\pi_m) \leq \frac{4}{3} \sum_{m \in [M]} E_m^l \mathbb{E}_{\Upsilon^{l-1}} \left[ \widehat{\text{Reg}}_m^l(\pi_m \mid \Upsilon^{l-1}) \right] + \frac{\sum_{m \in [M]} E_m^l K_m}{3c\gamma^l}
$$

$$+ \frac{25c^2 \sum_{m \in [M]} E_m^l K_m}{4\gamma^l}$$

$$\leq 2 \sum_{m \in [M]} E_m^l \mathbb{E}_{\Upsilon^{l-1}} \left[ \widehat{\text{Reg}}_m^l(\pi_m \mid \Upsilon^{l-1}) \right] + \eta^l$$

Also, it similarly holds that

$$\widehat{\text{Reg}}_m^l(\pi_m \mid \Upsilon^{l-1}) - \text{Reg}_m(\pi_m)$$
$$= \widehat{\mathcal{R}}_m^l(\widehat{\pi}_m^l \mid \Upsilon^{l-1}) - \widehat{\mathcal{R}}_m^l(\pi_m \mid \Upsilon^{l-1}) - (\mathcal{R}_m(\pi_m^*) - \mathcal{R}_m(\pi_m))$$
$$\leq \widehat{\mathcal{R}}_m^l(\widehat{\pi}_m^l \mid \Upsilon^{l-1}) - \widehat{\mathcal{R}}_m^l(\pi_m \mid \Upsilon^{l-1}) - (\mathcal{R}_m(\widehat{\pi}_m^l) - \mathcal{R}_m(\pi_m))$$
$$= \widehat{\mathcal{R}}_m^l(\widehat{\pi}_m^l \mid \Upsilon^{l-1}) - \mathcal{R}_m(\widehat{\pi}_m^l) + \mathcal{R}_m(\pi_m) - \widehat{\mathcal{R}}_m^l(\pi_m \mid \Upsilon^{l-1})$$
$$\leq \sqrt{V_m(p_m^{l-1}, \widehat{\pi}_m^l \mid \Upsilon^{l-1})} \sqrt{\mathcal{E}_m^{l-1}(\Upsilon^{l-1})} + \sqrt{V_m(p_m^{l-1}, \pi_m \mid \Upsilon^{l-1})} \sqrt{\mathcal{E}_m^{l-1}(\Upsilon^{l-1})}$$
$$\leq \frac{K_m + \gamma^{l-1} \widehat{\text{Reg}}_m^{l-1}(\widehat{\pi}_m^l \mid \Upsilon^{l-1})}{8c\gamma^l} + \frac{K_m + \gamma^{l-1} \widehat{\text{Reg}}_m^{l-1}(\pi_m \mid \Upsilon^{l-1})}{8c\gamma^l} + 4c\gamma^l \mathcal{E}_m^{l-1}(\Upsilon^{l-1}).$$

Then, summing over $M$ agents, we can obtain that

$$\mathbb{E}_{\Upsilon^{l-1}} \left[ \sum_{m \in [M]} E_m^l \left( \widehat{\text{Reg}}_m^l(\pi_m \mid \Upsilon^{l-1}) - \text{Reg}_m(\pi_m) \right) \right]$$

$$\leq \frac{\sum_{m \in [M]} E_m^l K_m}{4c\gamma^l} + \frac{\overline{c}\gamma^{l-1}}{8c\gamma^l} \sum_{m \in [M]} E_m^{l-1} \mathbb{E}_{\Upsilon^{l-1}} \left[ \widehat{\text{Reg}}_m^{l-1}(\widehat{\pi}_m^l \mid \Upsilon^{l-1}) \right]$$

$$+ \frac{\overline{c}\gamma^{l-1}}{8c\gamma^l} \sum_{m \in [M]} E_m^{l-1} \mathbb{E}_{\Upsilon^{l-1}} \left[ \widehat{\text{Reg}}_m^{l-1}(\pi_m \mid \Upsilon^{l-1}) \right] + 4c\gamma^l \sum_{m \in [M]} E_m^l \mathbb{E}_{\Upsilon^{l-1}} \left[ \mathcal{E}_m^{l-1}(\Upsilon^{l-1}) \right]$$

$$\leq \frac{\sum_{m \in [M]} E_m^l K_m}{4c\gamma^l} + \frac{\overline{c}\gamma^{l-1}}{4c\gamma^l} \sum_{m \in [M]} E_m^{l-1} \mathbb{E}_{\Upsilon^{l-1}} \left[ \text{Reg}_m(\widehat{\pi}_m^l \mid \Upsilon^{l-1}) \right]$$

$$+ \frac{\overline{c}\gamma^{l-1}}{4c\gamma^l} \sum_{m \in [M]} E_m^{l-1} \text{Reg}_m(\pi_m) + \frac{\overline{c}\gamma^{l-1}}{4c\gamma^l} \cdot \eta^{l-1} + 4c\gamma^l \sum_{m \in [M]} E_m^l \mathbb{E}_{\Upsilon^{l-1}} \left[ \mathcal{E}_m^{l-1}(\Upsilon^{l-1}) \right]$$

$$\overset{(g)}{\leq} \frac{\sum_{m \in [M]} E_m^l K_m}{4c\gamma^l} + \frac{\gamma^{l-1}}{4\gamma^l} \cdot \eta^l + \frac{\gamma^{l-1}}{4\gamma^l} \sum_{m \in [M]} E_m^l \text{Reg}_m(\pi_m)$$

$$+ \frac{\overline{c}\gamma^{l-1}}{4c\gamma^l} \cdot \eta^{l-1} + 4c\gamma^l \sum_{m \in [M]} E_m^l \mathbb{E}_{\Upsilon^{l-1}} \left[ \mathcal{E}_m^{l-1}(\Upsilon^{l-1}) \right]$$

$$\leq \frac{\sum_{m \in [M]} E_m^l K_m}{4c\gamma^l} + \frac{9c^2 \sum_{m \in [M]} E_m^l K_m}{4\gamma^l} + \frac{1}{4} \sum_{m \in [M]} E_m^l \text{Reg}_m(\pi_m)$$

$$+ \frac{9c^2 \sum_{m \in [M]} E_m^l K_m}{4\gamma^l} + \frac{4c^2 \sum_{m \in [M]} E_m^l K_m}{\gamma^l},$$

where inequality (g) is from the previous derivation that

$$\sum_{m \in [M]} E_m^{l-1} \text{Reg}_m(\widehat{\pi}_m^l \mid \Upsilon^{l-1}) \leq 2\underline{c} \sum_{m \in [M]} E_m^l \widehat{\text{Reg}}_m^l(\widehat{\pi}_m^l \mid \Upsilon^{l-1}) + \underline{c}\eta^l = \underline{c}\eta^l$$

Thus, it holds that

$$\sum_{m \in [M]} E_m^l \mathbb{E}_{\Upsilon^{l-1}} \left[ \widehat{\text{Reg}}_m^{l-1}(\widehat{\pi}_m^l \mid \Upsilon^{l-1}) \right] \leq \frac{5}{4} \sum_{m \in [M]} E_m^l \text{Reg}_m(\pi_m)$$

$$+ \frac{\sum_{m\in[M]} E_m^l K_m}{4c\gamma^l} + \frac{17c^2 \sum_{m\in[M]} E_m^l K_m}{2\gamma^l}$$

$$\Rightarrow \sum_{m\in[M]} E_m^l \mathbb{E}_{\Upsilon^{l-1}} \left[ \widehat{\text{Reg}}_m^{l-1}(\hat{\pi}_m^l \mid \Upsilon^{l-1}) \right] \le 2 \sum_{m\in[M]} E_m^l \text{Reg}_m(\pi_m) + \eta^l.$$

With these two parts, the lemma can be obtained by induction. $\qquad\square$

Furthermore, the following lemma provides a characterization of the per-epoch loss of the federation.

**Lemma C.5.** *For every epoch $l > 1$, conditioned on $\Upsilon^{l-1}$, it holds that*

$$\mathbb{E}_{\Upsilon^{l-1}} \left[ \sum_{m\in[M]} E_m^l \sum_{\pi_m\in\Psi_m} Q_m^l(\pi_m \mid \Upsilon^{l-1}) \text{Reg}_m(\pi_m) \right] \le \frac{11c^2}{\gamma^l} \sum_{m\in[M]} E_m^l K_m,$$

*where $Q^l(\cdot|\Upsilon^{l-1})$ is a probability measure on $\Psi_m$ defined in Lemma C.7*

*Proof.* For any probability measures $\{\tilde{Q}_m^l(\cdot) : m \in [M]\}$, where $\tilde{Q}_m^l(\cdot)$ is on $\Psi_M$, it holds that

$$\sum_{m\in[M]} E_m^l \sum_{\pi_m\in\Psi_m} \tilde{Q}_m^l(\pi_m)\text{Reg}_m(\pi_m)$$

$$\stackrel{(a)}{\le} 2\mathbb{E}_{\Upsilon^{l-1}} \left[ \sum_{\pi_{[M]}\in\Psi_{[M]}} \tilde{Q}^l(\pi_{[M]}) \sum_{m\in[M]} E_m^l \widehat{\text{Reg}}_m(\pi_m \mid \Upsilon^{l-1}) \right] + \eta^l$$

$$= 2\mathbb{E}_{\Upsilon^{l-1}} \left[ \sum_{m\in[M]} E_m^l \sum_{\pi_m\in\Psi_m} \tilde{Q}_m^l(\pi_m)\widehat{\text{Reg}}_m(\pi_m \mid \Upsilon^{l-1}) \right] + \eta^l,$$

where inequality (a) is from Lemma C.4 and $\tilde{Q}^l(\pi_{[M]}) := \prod_{m\in[M]} \tilde{Q}_m^l(\pi_m)$. Thus, we can obtain that

$$\mathbb{E}_{\Upsilon^{l-1}} \left[ \sum_{m\in[M]} E_m^l \sum_{\pi_m\in\Psi_m} Q_m^l(\pi_m \mid \Upsilon^{l-1})\text{Reg}_m(\pi_m) \right]$$

$$\le 2\mathbb{E}_{\Upsilon^{l-1}} \left[ \sum_{m\in[M]} E_m^l \sum_{\pi_m\in\Psi_m} Q_m^l(\pi_m \mid \Upsilon^{l-1})\widehat{\text{Reg}}_m(\pi_m \mid \Upsilon^{l-1}) \right] + \eta^l$$

$$\stackrel{(b)}{\le} \frac{2}{\gamma^l} \sum_{m\in[M]} E_m^l K_m + \frac{9c^2}{\gamma^l} \sum_{m\in[M]} E_m^l K_m$$

$$\le \frac{11c^2}{\gamma^l} \sum_{m\in[M]} E_m^l K_m,$$

where inequality (b) is from Lemma C.9. $\qquad\square$

With the previous lemmas, we can obtain the final Theorem 4.1, which is restated in the following.

**Theorem C.6** (Restatement of Theorem 4.1). *Using a learning rate*

$$\gamma^l = O\left( \sqrt{ \sum_{m\in[M]} E_m^{l-1} K_m \Big/ \left( \sum_{m\in[M]} E_m^{l-1} \mathcal{E}(E_{[M]}^{l-1}) \right) } \right)$$

*in epoch $l$, denoting $\bar{K}^l := \sum_{m\in[M]} E_m^l K_m / \sum_{m\in[M]} E_m^l$, the regret of FedIGW can be bounded as*

$$\text{Reg}(T) = O\left( \sum_{m\in[M]} E_m^1 + \sum_{l\in[2,l(T)]} c^{\frac{5}{2}} \sqrt{\bar{K}^l \mathcal{E}(E_{[M]}^{l-1})} \sum_{m\in[M]} E_m^l \right).$$

*Proof of Theorem 4.1.* The expected regret can be bounded as

$$\mathrm{Reg}(T) = \mathbb{E}\left[\sum_{m\in[M]}\sum_{t_m\in[T_m]}(f^*(x_{m,t_m},\pi_m^*(x_{m,t_m})) - f^*(x_{m,t_m},a_{m,t_m}))\right]$$

$$\leq \mathbb{E}\left[\sum_{l\in[2,l(T)]}\sum_{m\in[M]}\sum_{t_m\in[t_m(\tau^{l-1})+1,t_m(\tau^l)]}(f^*(x_{m,t_m},\pi_m^*(x_{m,t_m})) - f^*(x_{m,t_m},a_{m,t_m}))\right] + \sum_{m\in[M]}E_m^1$$

$$= \sum_{l\in[2,l(T)]}\mathbb{E}_{\Upsilon^{l-1}}\left[\mathbb{E}_{x_m,a_m^l}\left[\sum_{m\in[M]}E_m^l(f^*(x_m,\pi_m^*(x_m)) - f^*(x_m,a_m))\mid\Upsilon^{l-1}\right]\mid\Upsilon^{l-1}\right] + \sum_{m\in[M]}E_m^1$$

$$\overset{(a)}{=} \sum_{l\in[2,l(T)]}\mathbb{E}_{\Upsilon^{l-1}}\left[\sum_{m\in[M]}E_m^l\sum_{\pi_m\in\Psi^m}Q_m^l(\pi_m\mid\Upsilon^{l-1})\mathrm{Reg}_m(\pi_m)\mid\Upsilon^{l-1}\right] + \sum_{m\in[M]}E_m^1$$

$$\overset{(b)}{\leq} \sum_{l\in[2,l(T)]}\frac{11c^2}{\gamma^l}\sum_{m\in[M]}E_m^lK_m + \sum_{m\in[M]}E_m^1$$

$$\overset{(c)}{=} \sum_{l\in[2,l(T)]}11c^2\sqrt{\frac{\sum_{m\in[M]}E_m^{l-1}\mathcal{E}(\mathcal{F};E_{[M]}^{l-1})}{\sum_{m\in[M]}E_m^{l-1}K_m}}\sum_{m\in[M]}E_m^lK_m + \sum_{m\in[M]}E_m^1$$

$$\leq \sum_{l\in[2,l(T)]}11c^2\sqrt{\overline{K}\mathcal{E}(\mathcal{F};E_{[M]}^{l-1})}\sum_{m\in[M]}E_m^{l-1} + \sum_{m\in[M]}E_m^1,$$

where equality (a) is from Lemma C.8, inequality (b) is from Lemma C.5, and inequality (c) is from the choice of $\gamma^l$. The proof is then concluded. □

## C.3 Supporting Lemmas

The following supporting lemmas can be similarly obtained by the corresponding proofs in Simchi-Levi & Xu (2022).

**Lemma C.7** (Lemma 3, Simchi-Levi & Xu (2022)). *For any epoch $l\in\mathbb{N}$, conditioned on $\Upsilon^{l-1}$, there exists a probability measure $Q_m^l(\cdot|\Upsilon^{l-1})$ on $\Psi_m$ such that*

$$\forall a_m\in\mathcal{A}_m, \forall x_m\in\mathcal{X}_m,\qquad p_m^l(a_m|x_m,\Upsilon^{l-1}) = \sum_{\pi_m\in\Psi_m}\mathbb{1}\{\pi_m(x_m)=a_m\}Q_m^l(\pi_m|\Upsilon^{l-1}).$$

**Lemma C.8** (Lemma 4, Simchi-Levi & Xu (2022)). *Fix any epoch $l\in\mathbb{N}$, we have*

$$\mathbb{E}_{x_m\sim\mathcal{D}_m^{\mathcal{X}_m},a_m^l\sim p_m^l(\cdot|x_m)}\left[f^*(x_m,\pi_m^*(x_m)) - f^*(x_m,a_m^l)\mid\Upsilon^{l-1}\right]$$
$$= \sum_{\pi_m\in\Psi_m}Q_m^l(\pi_m\mid\Upsilon^{l-1})\mathrm{Reg}_m(\pi_m).$$

**Lemma C.9** (Lemma 5, Simchi-Levi & Xu (2022)). *Fix any epoch $l\in\mathbb{N}$, conditioned on $\Upsilon^{l-1}$, we have*

$$\sum_{\pi\in\Psi_m}Q_m^l(\pi_m\mid\Upsilon^{l-1})\widehat{\mathrm{Reg}}_m^l(\pi_m\mid\Upsilon^{l-1}) \leq \frac{K_m}{\gamma^l}.$$

**Lemma C.10** (Lemma 6, Simchi-Levi & Xu (2022)). *Fix any epoch $l\in\mathbb{N}$, for any policy $\pi_m\in\Psi_m$, we have*

$$V_m(p_m^l,\pi_m\mid\Upsilon^{l-1}) \leq K_m + \gamma^l\widehat{\mathrm{Reg}}_m^l(\pi_m\mid\Upsilon^{l-1}).$$

# D Proofs for Section 4.2

## D.1 Proofs of Corollary 4.2

First, with realizability, i.e., Assumption 3.1, the following characterization can be obtained.

**Lemma D.1** (Lemma 4.2, Agarwal et al. (2012))**.** *Fix a function $f \in \mathcal{F}$. Suppose we sample $x_m, r_m$ from the data distribution $\mathcal{D}_m$, and an action $a_m$ from an arbitrary distribution such that $r_m$ and $a_m$ are conditionally independent given $x_m$. Define the random variable*

$$\ell_m(f) := (f(x_m, a_m) - r_m(a_m))^2 - (f^*(x_m, a_m) - r_m(a_m))^2.$$

*Then, we have*

$$\mathbb{E}_{x_m, r_m, a_m}\left[\ell_m(f)\right] = \mathbb{E}_{x_m, a_m}\left[(f(x_m, a_m) - f^*(x_m, a_m))^2\right]$$

*and*

$$\mathbb{V}_{x_m, r_m, a_m}\left[\ell_m(f)\right] \leq 4\mathbb{E}_{x_m, r_m, a_m}\left[\ell_m(f)\right],$$

*where $\mathbb{V}[\cdot]$ denotes the variance of a random variable.*

Then, we establish an upper bound for the excess risk bound required in Definition C.1 via the following lemma

**Lemma D.2.** *Under the setup of Assumption C.1, if the adopted FL protocol provides an exact minimizer for the optimization problem in Eqn. (1) with quadratic losses, i.e.,*

$$\widehat{f} = \arg\min_{f \in \mathcal{F}} \frac{1}{n} \sum_{m \in [M]} \sum_{i \in [n_m]} \left(f(x_m^i, a_m^i) - y_m^i\right)^2,$$

*then, with probability at least $1 - \delta$, it holds that*

$$\sum_{m \in [M]} \frac{n_m}{n} \cdot \mathbb{E}_{x_m \sim \mathcal{D}_m^{x_m}, a_m \sim p_m(\cdot | x_m)} \left[\left(\widehat{f}(x_m, a_m) - f^*(x_m, a_m)\right)^2\right] \leq \frac{25 \log(|\mathcal{F}|/\delta)}{n}.$$

*As a result, Definition C.1 holds with*

$$\mathcal{E}(\delta, n_{[M]}) \leq O\left(\log(|\mathcal{F}|n)/n\right).$$

*Proof.* For simplicity, we abbreviate the quadratic loss associated with a fixed function $f \in \mathcal{F}$ as

$$\ell_m^i(f) = \ell_m(f(x_m^i, a_m^i); r_m^i) := \left(f(x_m^i, a_m^i) - r_m^i\right)^2, \qquad \forall m \in [M].$$

Then, with a probability at least $1 - \delta$, for a fixed $f \in \mathcal{F}$, it holds that

$$\sum_{m \in [M]} \sum_{i_m \in [n_m]} \mathbb{E}_{x_m^i, r_m^i, a_m^i}\left[\ell_m^i(f) - \ell_m^i(f^*)\right] - \sum_{m \in [M]} \sum_{i \in [n_m]} \left[\ell_m^i(f) - \ell_m^i(f^*)\right]$$

$$\overset{(a)}{\leq} 2\sqrt{\sum_{m \in [M]} \sum_{i_m \in [n_m]} \mathbb{V}_{x_m^i, r_m^i, a_m^i}\left[\ell_m^i(f) - \ell_m^i(f^*)\right] \log(1/\delta)} + \frac{4}{3}\log(1/\delta)$$

$$\overset{(b)}{\leq} 4\sqrt{\sum_{m \in [M]} \sum_{i_m \in [n_m]} \mathbb{E}_{x_m^i, r_m^i, a_m^i}\left[\ell_m^i(f) - \ell_m^i(f^*)\right] \log(1/\delta)} + \frac{4}{3}\log(1/\delta),$$

where inequality (a) leverages Bernstein's inequality and inequality (b) is based on Lemma D.1.

With

$$X(f) = \sqrt{\sum_{m \in [M]} \sum_{i_m \in [n_m]} \mathbb{E}_{x_m^i, r_m^i, a_m^i}\left[\ell_m^i(f) - \ell_{m,i}(f^*)\right]};$$

$$Z(f) = \sum_{m \in [M]} \sum_{i \in [n_m]} \left[\ell_m^i(f) - \ell_{m,i}(f^*)\right]; \qquad C = \sqrt{\log(1/\delta)},$$

applying a union bound to the above inequality indicates that with probability $1 - |\mathcal{F}|\delta$, for all $f \in \mathcal{F}$, it holds that

$$X(f)^2 - Z(f) \leq 4CX(f) + \frac{4}{3}C^2 \quad \Rightarrow \quad (X(f) - 2C)^2 - Z(f) \leq \frac{16}{3}C^2.$$

Since $\widehat{f}$ satisfies that $Z(\widehat{f}) \leq 0$, we can obtain that

$$X(\widehat{f})^2 \leq 25C^2,$$

In other words, with probability $1 - \delta$, it holds that

$$\sum_{m \in [M]} \sum_{i_m \in [n_m]} \mathbb{E}_{x_m^i, r_m^i, a_m^i} \left[ \left( \widehat{f}(x_m^i, a_m^i) - r_m^i \right)^2 - \left( f^*(x_m^i, a_m^i) - r_m^i \right)^2 \right]$$

$$= \sum_{m \in [M]} n_m \mathbb{E}_{x_m^i, a_m^i} \left[ \left( \widehat{f}(x_m^i, a_m^i) - f^*(x_m^i, a_m^i) \right)^2 \right] \leq 25 \log(|\mathcal{F}|/\delta),$$

where the equality is from the realizability in Assumption 3.1. The first half of the lemma is then proved.

With $\delta = 1/n$, the second half can be obtained as

$$\mathbb{E}_{S_{[M]}} \left[ \sum_{m \in [M]} \frac{n_m}{n} \cdot \mathbb{E}_{x_m, a_m} \left[ \left( \widehat{f}(x_m, a_m) - f^*(x_m, a_m) \right)^2 \right] \right] \leq \frac{25 \log(|\mathcal{F}|n)}{n} + \frac{1}{n},$$

which concludes the proof. $\qquad\square$

Based on the established excess risk bound, Corollary 4.2 can be obtained as follows.

**Corollary D.3** (Restatement of Corollary 4.2). *If $|\mathcal{F}| < \infty$ and the adopted FL protocol provides an exact minimizer for Eqn.* (1) *with quadratic losses, with $\tau^l = 2^l$, FedIGW incurs a regret of*

$$\text{Reg}(T) = O(\sqrt{KMT \log(|\mathcal{F}|MT)})$$

*and a total $O(\log(T))$ calls of the adopted FL protocol.*

*Proof of Corollary 4.2.* With Theorem 4.1 and Lemma D.2, under the choice of $\tau^l = 2^l$, the regret can be bounded as

$$\text{Reg}(T) = O \left( ME^1 + \sum_{l \in [2, l(T)]} \sqrt{KME^l \log(|\mathcal{F}|ME^l)} \right)$$

$$= O \left( \sum_{l \in [2, \lceil \log_2(T) \rceil]} \sqrt{KM2^l \log(|\mathcal{F}|MT)} \right)$$

$$= O \left( \sqrt{KMT \log(|\mathcal{F}|MT)} \right),$$

and the exponentially growing epoch length naturally leads to $O(\log(T))$ calls of the adopted FL protocol, which concludes the proof. $\qquad\square$

### D.2 Proofs of Corollary 4.4 and Additional Results

In the following, we first prove Lemma 4.3 while also noting that this result is general and does not rely on the specific parameterization of $\mathcal{F}$, although we presented it with the $d$-dimensional parameterization considered in Section 4.2.

**Lemma D.4** (Complete Version of Lemma 4.3). *If the loss function $l_m(\cdot; \cdot)$ is $\mu_f$-strongly convex in its first coordinate for all $m \in [M]$, i.e.,*

$$l_m(z_1'; z_2) - l_m(z_1; z_2) \geq \frac{dl_m(z_1; z_2)}{dz_1} \cdot (z_1' - z_1) + \frac{\mu_f}{2}(z_1' - z_1)^2, \quad \text{for any } z_1, z_1' \text{ and } z_2,$$

*and*

$$\inf_{y \in \mathbb{R}} \mathbb{E}_{r_m}[l_m(y, r_m(a_m))|x_m, a_m] = \mathbb{E}_{r_m}[l(f_{\omega^*}(x_m, a_m), r_m(a_m))|x_m, a_m] \qquad (3)$$

*for all $m \in [M]$, $(x_m, a_m) \in \mathcal{X}_m \times \mathcal{A}_m$, Definition C.1 holds with*

$$\mathcal{E}(\mathcal{F}; n_{[M]}) \geq 2 \left( \varepsilon_{opt}(\mathcal{F}; n_{[M]}) + \varepsilon_{gen}(\mathcal{F}; n_{[M]}) \right) / \mu_f,$$

*where*

$$\varepsilon_{gen}(\mathcal{F}; n_{[M]}) := \mathbb{E}_{\mathcal{S}, \xi}[\mathcal{L}(f_{\widehat{\omega}_{\mathcal{S}}}) - \widehat{\mathcal{L}}(f_{\widehat{\omega}_{\mathcal{S}}}; \mathcal{S})];$$
$$\varepsilon_{opt}(\mathcal{F}; n_{[M]}) := \mathbb{E}_{\mathcal{S}, \xi}[\widehat{\mathcal{L}}(f_{\widehat{\omega}_{\mathcal{S}}}; \mathcal{S}) - \widehat{\mathcal{L}}(f_{\omega_{\mathcal{S}}^*}; \mathcal{S})].$$

*Proof.* First, for any $\widehat{\omega}_{\mathcal{S}}$, it holds that

$$\mathcal{L}(f_{\widehat{\omega}_{\mathcal{S}}}) - \mathcal{L}(f_{\omega^*})$$
$$= \sum_{m \in [M]} \frac{n_m}{n} \mathbb{E}_{x_{m,i}, a_{m,i}, r_{m,i}} \left[ \ell(f_{\widehat{\omega}_{\mathcal{S}}}(x_{m,i}, a_{m,i}); r_{m,i}) - \ell(f_{\omega^*}(x_{m,i}, a_{m,i}); r_{m,i}) \right]$$
$$\geq \frac{\mu_f}{2} \sum_{m \in [M]} \frac{n_m}{n} \mathbb{E}_{x_{m,i}, a_{m,i}} \left[ (f_{\widehat{\omega}_{\mathcal{S}}}(x_{m,i}, a_{m,i}) - f_{\omega^*}(x_{m,i}, a_{m,i}))^2 \right]$$

where the inequality is due to the strong convexity of $\ell(\cdot; \cdot)$ w.r.t. its first coordinate and the optimality of $f_{\omega^*}$ assumed in Eqn. (3). Thus, we obtain that

$$\sum_{m \in [M]} \frac{n_m}{n} \mathbb{E}_{x_{m,i}, a_{m,i}} \left[ (f_{\widehat{\omega}_{\mathcal{S}}}(x_{m,i}, a_{m,i}) - f_{\omega^*}(x_{m,i}, a_{m,i}))^2 \right] \leq \frac{2}{\mu_f} \left( \mathcal{L}(f_{\widehat{\omega}_{\mathcal{S}}}) - \mathcal{L}(f_{\omega^*}) \right).$$

Furthermore, it holds that

$$\mathbb{E}_{\mathcal{S}, \xi} \left[ \mathcal{L}(f_{\widehat{\omega}_{\mathcal{S}}}) \right] - \mathcal{L}(f_{\omega^*})$$
$$= \mathbb{E}_{\mathcal{S}, \xi} \left[ \mathcal{L}(f_{\widehat{\omega}_{\mathcal{S}}}) \right] - \mathbb{E}_{\mathcal{S}, \xi} \left[ \widehat{\mathcal{L}}(f_{\widehat{\omega}_{\mathcal{S}}}; \mathcal{S}) \right] + \mathbb{E}_{\mathcal{S}, \xi} \left[ \widehat{\mathcal{L}}(f_{\widehat{\omega}_{\mathcal{S}}}; \mathcal{S}) \right] - \mathcal{L}(f_{\omega^*})$$
$$\leq \mathbb{E}_{\mathcal{S}, \xi} \left[ \mathcal{L}(f_{\widehat{\omega}_{\mathcal{S}}}) \right] - \mathbb{E}_{\mathcal{S}, \xi} \left[ \widehat{\mathcal{L}}(f_{\widehat{\omega}_{\mathcal{S}}}; \mathcal{S}) \right] + \mathbb{E}_{\mathcal{S}, \xi} \left[ \widehat{\mathcal{L}}(f_{\widehat{\omega}_{\mathcal{S}}}; \mathcal{S}) \right] - \mathbb{E}_{\mathcal{S}, \xi} \left[ \widehat{\mathcal{L}}(f_{\omega_{\mathcal{S}}^*}; \mathcal{S}) \right],$$

where the last inequality is due to

$$\mathcal{L}(f_{\omega^*}) = \mathbb{E}_{\mathcal{S}} \left[ \widehat{\mathcal{L}}(f_{\omega^*}; \mathcal{S}) \right] \geq \mathbb{E}_{\mathcal{S}} \left[ \widehat{\mathcal{L}}(f_{\omega_{\mathcal{S}}^*}; \mathcal{S}) \right].$$

The proof is then concluded. $\square$

Then, for the generalization error analyses, the following lemma can be obtained via standard proofs (e.g., Theorem 6.4 in Zhang (2023); Theorem 3.3 in Mohri et al. (2018)).

**Lemma D.5.** *It holds that*

$$\varepsilon_{gen}(\mathcal{F}; n_{[M]}) := \mathbb{E}_{\mathcal{S}, \xi}[\mathcal{L}(f_{\widehat{\omega}_{\mathcal{S}}}) - \widehat{\mathcal{L}}(f_{\widehat{\omega}_{\mathcal{S}}}; \mathcal{S})] \leq 2\Re(\mathcal{F}; n_{[M]}).$$

*Here, the distributional-independent upper bound $\Re(\mathcal{F}; n_{[M]})$ on the Rademacher complexity is defined as*

$$\Re(\mathcal{F}; n_{[M]}) := \sup \left\{ \mathbb{E}_{\mathcal{S}_{[M]}, \sigma} \left[ \sup_{\omega} \left\{ \sum_{m \in [M]} \frac{1}{n} \sum_{i \in [n_m]} \sigma_{m,i} \cdot \ell_m(f_\omega(x_{m,i}, a_{m,i}); r_{m,i}) \right\} \right] \right\}, \quad (4)$$

*where the outside supremum is over possible distributions of dataset $\mathcal{S}$ defined in Definition C.1 and the expectation is w.r.t. the generation of dataset $\mathcal{S}_{[M]}$ following a fixed distribution and independent Rademacher random variables $\sigma := \{\sigma_{m,i} : m \in [M], i \in [n_m]\}$.*

The optimization error of FedAvg (McMahan et al., 2017) and SCAFFOLD (Karimireddy et al., 2020) are presented in Appendices F.1 and F.2. Combining the generalization error and optimization error via Lemma 4.3 into Theorem 4.1, Corollary 4.4 can be obtained, which is restated in the following.

**Corollary D.6** (Restatement of Corollary 4.4). *Under the condition of Lemma 4.3, the regret of FedIGW can be bounded as*

$$\text{Reg}(T) = O\left( ME^1 + \sum\nolimits_{l \in [2, l(T)]} \sqrt{K\left(\mathfrak{R}^{l-1} + \varepsilon_{opt}^l\right) / \mu_f} ME^l \right),$$

*where $\mathfrak{R}^l := \mathfrak{R}(\mathcal{F}; \{E^l : m \in [M]\})$ and using $\rho^l$ rounds of agents-server communications (i.e., global aggregations) and $\kappa^l$ rounds of local updates in epoch $l$, under certain assumptions,*

- *with **FedAvg** as $\texttt{FLroutine}(\cdot)$, if $\widehat{\mathcal{L}}_m(f_\omega; \mathcal{S}_{[M]}^l)$ is $\mu_\omega$-strongly convex and $\beta_\omega$-smooth w.r.t. $\omega$ for all $m \in [M]$ while the gradients are unbiased, have a $\sigma_b^2$-bounded variance and have a $G_b$-bounded dissimilarity, the output $f_{\widehat{\omega}^l}$ satisfies that $\varepsilon_{opt}^l := \varepsilon_{opt}(\mathcal{F}; n_{[M]}^l) \le \tilde{O}(\sigma_b^2(\mu_\omega \rho^l \kappa^l M)^{-1} + \beta_\omega G_b^2(\mu_\omega \rho^l)^{-2})$, when $\rho^l \ge \Omega(\beta_\omega / \mu_\omega)$ (see Lemma F.1 for the full statement);*

- *with **SCAFFOLD** as $\texttt{FLroutine}(\cdot)$, if $\widehat{\mathcal{L}}_m(f_\omega; \mathcal{S}_{[M]}^l)$ is $\mu_\omega$-strongly convex and $\beta_\omega$-smooth w.r.t. $\omega$ for all $m \in [M]$ while the gradients are unbiased and have a $\sigma_b^2$-bounded variance, the output $f_{\widehat{\omega}^l}$ satisfies that $\varepsilon_{opt}^l := \varepsilon_{opt}(\mathcal{F}; n_{[M]}^l) \le \tilde{O}(\sigma_b^2(\mu_\omega \mathfrak{R}^l \kappa^l M)^{-1})$, when $\rho^l \ge \Omega(\beta_\omega / \mu_\omega)$ (see Lemma F.6 for the full statement);.*

By further setting a suitable number of global aggregations for each epoch such that the optimization error is on the same order as the generalization error, the following more specific corollary can obtained for FedAvg and SCAFFOLD, which can be easily extended for other FL designs.

**Corollary D.7.** *Under the conditions of Lemma 4.3 and Corollary D.6, FedIGW incurs a regret of*

$$\text{Reg}(T) = O\left( ME^1 + \sum\nolimits_{l \in [2, l(T)]} \sqrt{K\mathfrak{R}^{l-1}/\mu_f} ME^l \right)$$

*with the following bounds on the rounds of communications*

$$\tilde{O}\left( \sum_{l \in [l(T)]} \frac{\beta_\omega}{\mu_\omega} + \frac{\sigma_b^2}{\mu_\omega \mathfrak{R}^l \kappa^l M} + \sqrt{\frac{\beta_\omega G_b^2}{\mu_\omega^2 \mathfrak{R}^l}} \right) \qquad \text{(using FedAvg)};$$

$$\tilde{O}\left( \sum_{l \in [l(T)]} \frac{\beta_\omega}{\mu_\omega} + \frac{\sigma_b^2}{\mu_\omega \mathfrak{R}^l \kappa^l M} \right) \qquad \text{(using SCAFFOLD)},$$

*where $\mathfrak{R}^l := \mathfrak{R}(\mathcal{F}_{[M]}, \{E^l : m \in [M]\})$ and $\kappa^l$ is the number of local updates in epoch $l$.*

*Proof.* From Corollary D.6, when using FedAvg as the adopted FL protocol in FedIGW, the optimization error in epoch $l$ of form

$$\tilde{O}\left( \frac{\sigma_b^2}{\mu_\omega \rho^l \kappa^l M} + \frac{\beta_\omega G_b^2}{\mu_\omega^2 (\rho^l)^2} \right),$$

when $\rho^l = \Omega(\beta_\omega / \mu_\omega)$. Thus, if the communication rounds

$$\rho^l = \tilde{\Theta}\left( \frac{\beta_\omega}{\mu_\omega} + \frac{\sigma_b^2}{\mu_\omega \mathfrak{R}^l \kappa^l M} + \sqrt{\frac{\beta_\omega G_b^2}{\mu_\omega^2 \mathfrak{R}^l}} \right).$$

we are guaranteed to have the optimization error on the order of $O(\mathfrak{R}^l)$.

Then, the regret in Corollary 4.4 is of order

$$\text{Reg}(T) = O\left( ME^1 + \sum\nolimits_{l \in [2, l(T)]} \sqrt{K\mathfrak{R}^{l-1}/\mu_f} ME^l \right)$$

while the overall communication rounds can be bounded as

$$\sum_{l \in [l(T)]} \rho^l = \tilde{O}\left( \sum_{l \in [l(T)]} \frac{\beta_\omega}{\mu_\omega} + \frac{\sigma_b^2}{\mu_\omega \mathfrak{R}^l \kappa^l M} + \sqrt{\frac{\beta_\omega G_b^2}{\mu_\omega^2 \mathfrak{R}^l}} \right),$$

which concludes the proof for FedAvg. The result of using SCAFFOLD can be similarly obtained. $\square$

### D.3 A Linear Reward Function Class

We here provide a detailed discussion on the linear reward function class considered in Remark 4.5 at the end of Section 4.2. Especially, following standard assumptions in linear bandits (Abbasi-Yadkori et al., 2011) and federated linear bandits (Li & Wang, 2022a; He et al., 2022; Amani et al., 2022), we consider $\mu_m(x_m, a_m) = \langle \phi(x_m, a_m), \omega^* \rangle$, where $\phi(\cdot)$ is a known $d$-dimensional mapping and $\omega^*$ is an unknown $d$-dimensional system parameter. Then, it is sufficient to consider a linear function class $\mathcal{F}$, where $f_\omega(\cdot) = \langle \omega, \phi(\cdot) \rangle$ and $f^*(\cdot) = \langle \omega^*, \phi(\cdot) \rangle$. Moreover, for convenience, we assume that $\|\phi(x_m, a_m)\|_2 \leq 1$ and $\|\omega^*\|_2 \leq 1$.

As mentioned in Remark 4.5, the FL problem can be formulated as a standard ridge regression with

$$\ell_m(f_\omega(x_m, a_m); r_m) := (\langle \omega, \phi(x_m, a_m) \rangle - r_m)^2 + \lambda \|\omega\|_2^2.$$

In other words, Eqn. (1) can be restated as

$$\min_{\omega \in \mathbb{R}^d} \widehat{\mathcal{L}}(f_\omega; \mathcal{S}) := \sum_{m \in [M]} \frac{1}{n} \sum_{i \in [n_m]} \left( \langle \omega, \phi(x_m^i, a_m^i) \rangle - r_m^i \right)^2 + \lambda \|\omega\|_2^2, \tag{5}$$

which has an exact minimizer as

$$\omega_{\mathcal{S}}^* = \left( \frac{1}{n} \sum_{m \in [M]} \sum_{i \in [n_m]} \phi(x_m^i, a_m^i) \phi(x_m^i, a_m^i)^\top + \lambda I \right)^{-1} \left( \frac{1}{n} \sum_{m \in [M]} \sum_{i \in [n_m]} \phi(x_m^i, a_m^i) r_m^i \right). \tag{6}$$

We provide an excess risk bound required in Definition C.1 through the following decomposition:

$$\mathbb{E}_{\mathcal{S},\xi} \left[ \sum_{m \in [M]} \frac{n_m}{n} \mathbb{E}_{x_m, a_m} \left( \langle \widehat{\omega}_{\mathcal{S}}, \phi(x_m, a_m) \rangle - \langle \omega^*, \phi(x_m, a_m) \rangle \right)^2 \right]$$

$$\leq 2 \mathbb{E}_{\mathcal{S},\xi} \left[ \sum_{m \in [M]} \frac{n_m}{n} \mathbb{E}_{x_m, a_m} \left( \langle \widehat{\omega}_{\mathcal{S}}, \phi(x_m, a_m) \rangle - \langle \omega_{\mathcal{S}}^*, \phi(x_m, a_m) \rangle \right)^2 \right]$$

$$+ 2 \mathbb{E}_{\mathcal{S},\xi} \left[ \sum_{m \in [M]} \frac{n_m}{n} \mathbb{E}_{x_m, a_m} \left( \langle \omega_{\mathcal{S}}^*, \phi(x_m, a_m) \rangle - \langle \omega^*, \phi(x_m, a_m) \rangle \right)^2 \right]$$

$$= 2 \mathbb{E}_{\mathcal{S},\xi} \left[ \|\widehat{\omega}_{\mathcal{S}} - \omega_{\mathcal{S}}^*\|_\Sigma^2 \right]$$

$$+ 2 \mathbb{E}_{\mathcal{S}} \left[ \sum_{m \in [M]} \frac{n_m}{n} \mathbb{E}_{x_m, a_m} \left( \langle \omega_{\mathcal{S}}^*, \phi(x_m, a_m) \rangle - \langle \omega^*, \phi(x_m, a_m) \rangle \right)^2 \right]$$

$$\leq 2 \mathbb{E}_{\mathcal{S},\xi} \left[ \lambda_{\max}(\Sigma) \|\widehat{\omega}_{\mathcal{S}} - \omega_{\mathcal{S}}^*\|_2^2 \right] \qquad =: \text{term (A)}$$

$$+ 2 \mathbb{E}_{\mathcal{S}} \left[ \sum_{m \in [M]} \frac{n_m}{n} \mathbb{E}_{x_m, a_m} \left( \langle \omega_{\mathcal{S}}^*, \phi(x_m, a_m) \rangle - \langle \omega^*, \phi(x_m, a_m) \rangle \right)^2 \right] \qquad =: \text{term (B)}$$

where

$$\Sigma := \sum_{m \in [M]} \frac{n_m}{n} \mathbb{E}_{x_m, a_m} \left[ \phi(x_m, a_m) \phi(x_m, a_m)^\top \right]$$

and $\lambda_{\max}(\Sigma)$ denotes the maximum eigenvalue of $\Sigma$. With $\|\phi(x, a)\|_2 \leq 1$, it can be verified that $\lambda_{\max}(\Sigma) \leq 1$. In the above decomposition, term (A) can be interpreted as the optimization error, while term (B) is the generalization error.

We can then plug in the aforementioned explicit formula of $\omega_{\mathcal{S}}^*$ into term (B) and demonstrate that term (B) $= \widetilde{O}(d/n)$ with $\lambda = 1/n$ under the assumption that $\|\omega^*\|_2 \leq 1$ and $r_m \in [0, 1]$ (e.g., following Theorem 9.35 in Zhang (2023)).

For the ridge regression problem in Eqn. (5), previous designs on federated linear bandits typically (Wang et al., 2019; Dubey & Pentland, 2020; Li & Wang, 2022a; He et al., 2022; Amani et al., 2022) have agents collaboratively provide the exact minimizer in Eqn. (6) via directly communicating their local rewards aggregates, i.e., $\sum_{i \in [n_m]} \phi(x_m^i, a_m^i) r_m^i$, and local covariance matrices, i.e., $\sum_{i \in [n_m]} \phi(x_m^i, a_m^i) \phi(x_m^i, a_m^i)^\top$. Thus, one round of agent-server communication is sufficient, where $O(Md^2)$ real numbers are shared. However, directly sharing such compressed data is often undesired in FL studies due to privacy concerns. We refer to this protocol as the "**direct method**" for simplicity in the following discussions.

With the flexible FL choice in FedIGW, it can accommodate many other efficient optimization algorithms. In particular, a distributed version of accelerated gradient descent (AGD) (Nesterov, 2003) takes only $O(\sqrt{\kappa} \log(1/\varepsilon'))$ rounds of communications of gradients to have an optimization error of $\varepsilon'$, where $\kappa$ is the condition number (i.e., the ratio between the smooth and strongly convex parameter in the considered problem). With $\lambda = 1/n$, it holds that $\kappa = O(n)$; thus $O(\sqrt{n} \log(d/n))$ rounds of communications of gradients are sufficient to obtain an optimization error of order $\tilde{O}(d/n)$, where each agents' gradients are intuitively $d$-dimensional.

With the above illustration, the following corollary regarding the performance FedIGW with a linear reward function class is then a straightforward extension from Theorem 4.1.

**Corollary D.8.** *In the considered linear reward function class with shared true parameters, using the direct method or distributed AGD as the adopted FL protocol to solve the FL problem in Eqn.* (5) *and $\tau^l = 2^l$, FedIGW obtains a regret of*

$$\text{Reg}(T) = \tilde{O}\left(\sum_{l \in [\log_2(T)]} \sqrt{\frac{Kd}{M2^{l-1}}} M2^l\right) = \tilde{O}\left(\sqrt{MKdT}\right)$$

*and the amount of real numbers communicated can be bounded as*

$$O\left(\sum_{l \in [\log_2(T)]} Md^2\right) = O(Md^2 \log(T)) \qquad \text{(using the direct method)};$$

$$O\left(\sum_{l \in [\log_2(T)]} Md\sqrt{M2^l} \log(d/(M2^l))\right) = O(d\log(d)\sqrt{M^3T}) \qquad \text{(using distributed AGD)}.$$

# E    DETAILS OF SECTION 6

## E.1    PERSONALIZED LEARNING: DETAILS OF SECTION 6.1

Additional details for the personalized learning setting in Section 6.1 are discussed here. In particular, the overall algorithm structure still follows Algorithm 1, while the major difference is that a personalized FL problem is considered:

$$\min_{\omega^\alpha, \omega^\beta_{[M]}} \widehat{\mathcal{L}}(f_{\omega^\alpha, \omega^\beta_{[M]}}; \mathcal{S}_{[M]}) := \sum_{m \in [M]} \frac{n_m}{n} \widehat{\mathcal{L}}_m(f_{\omega^\alpha, \omega^\beta_m}; \mathcal{S}_m),$$

where

$$\widehat{\mathcal{L}}_m(f_{\omega^\alpha, \omega^\beta_m}; \mathcal{S}_m) := \frac{1}{n_m} \sum_{i \in [n_m]} \ell_m(f_{\omega^\alpha, \omega^\beta_m}(x_m^i, a_m^i); r_m^i).$$

Furthermore, to bound the generalization error, similar to the Rademacher complexity in Eqn. (4), a slightly different Rademacher complexity is introduced as

$$\mathfrak{P}(\mathcal{F}_{[M]}; n_{[M]}) = \sup\left\{\mathbb{E}_{\mathcal{S}, \boldsymbol{\sigma}}\left[\sup_{\omega^\alpha, \omega^\beta_{[M]}} \left\{\sum_{m \in [M]} \frac{1}{n} \sum_{i \in [n_m]} \sigma_{m,i} \cdot \ell_m(f_{\omega_m}(x_m^i, a_m^i); r_m^i)\right\}\right]\right\},$$

which is suitable for the considered personalized setting with parameters $[\omega^\alpha, \omega^\beta_{[M]}]$ involved. A similar notation is also adopted in Mohri et al. (2019).

The following corollary can then be established for the personalized version of FedIGW with the LSGD-PFL algorithm (Hanzely et al., 2021) adopted to solve the personalized FL task.

**Corollary E.1.** *Under the conditions of Lemmas 4.3 and F.7, with LSGD-PFL as the adopted personalized FL protocol, FedIGW incurs a regret of*

$$\text{Reg}(T) = O\left(ME^1 + \sum_{l \in [2,l(T)]} \sqrt{K\mathfrak{P}^{l-1}/\mu_f} ME^l\right)$$

*with*

$$\tilde{O}\left(\sum_{l \in [l(T)]} \max\{\beta_{\omega^\beta}(\kappa^l)^{-1}, \beta_{\omega^\alpha}\}\mu_\omega^{-1} + \sigma_b^2(\mu_\omega \kappa^l M\mathfrak{P}^l)^{-1} + \sqrt{\beta_{\omega^\alpha}(G^2 + \sigma^2)(\mu_\omega^2 \mathfrak{P}^l)^{-1}}\right)$$

*rounds of communications, where $\mathfrak{P}^l := \mathfrak{P}(\mathcal{F}_{[M]}, \{E^l : m \in [M]\})$ and $\kappa^l$ is the number of local updates in epoch $l$.*

The proof largely follows that of Corollary D.7: decomposing excess risk to generalization and optimization errors; using Rademacher complexity to characterize the generalization error; using FL convergence analyses to characterize the optimization error; and combining them together such that the optimization error does not dominate the generalization error. As the LSGD-PFL protocol (Hanzely et al., 2021) is adopted to solve the personalized FL task as an illustration, its corresponding convergence analyses should be incorporated, which is presented in Lemma F.7.

### E.1.1 A LINEAR REWARD FUNCTION CLASS

As an extension of the linear reward function in Appendix D.3, we consider that

$$\mu_m(x_m, a_m) = \langle \phi(x_m, a_m), \omega_m^* \rangle, \qquad \forall m \in [M], (x_m, a_m) \in \mathcal{X}_m \times \mathcal{A}_m,$$

and the true model parameters $\{\omega_m^* : m \in [M]\}$ follow Assumption 6.2, i.e., $\omega_m^* = [\omega^{\alpha,*}, \omega_m^{*,\beta}]$ with $\omega^{\alpha,*}$ shared among all agents.

It can be further realized that the above problem setting is identical to a $\tilde{d}$-dimensional linear system, where $\tilde{d} := d^\alpha + \sum_{m \in [M]} d_m^\beta$: the overall true model parameter is

$$\tilde{\omega}^* = \left[\omega^{*,\alpha}, \omega_1^{*,\beta}, \cdots, \omega_M^{*,\beta}\right] \in \mathbb{R}^{\tilde{d}}.$$

and a correspondingly feature mapping $\tilde{\phi}(\cdot)$ is

$$\tilde{\phi}(x_m, a_m) = \left[\phi(x_m, a_m)_{[1:d^\alpha]}, \boldsymbol{O}_{d_1^\beta}, \cdots, \boldsymbol{O}_{d_{m-1}^\beta}, \phi(x_m, a_m)_{[d^\alpha+1:d_m]}, \boldsymbol{O}_{d_{m+1}^\beta}, \cdots, \boldsymbol{O}_{d_M^\beta}\right],$$

i.e., an expanded version of the original feature, where $\phi(x_m, a_m)_{[i:j]} \in \mathbb{R}^{j-i+1}$ denotes the subvector containing $[i:j]$-th elements in $\phi(x_m, a_m)$ and $\boldsymbol{O}_i \in \mathbb{R}^i$ an $i$-dimensional null vector.

With this reformulated problem, discussions from Appendix D.3 can be directly leveraged. Especially, Corollary D.8 indicates the following result.

**Corollary E.2.** *In the considered linear reward function class with partially true parameters, using distributed AGD as the adopted FL protocol to solve the FL problem in Eqn. (5) with reformulated feature mapping $\tilde{\phi}(\cdot)$ and $\tau^l = 2^l$, FedIGW incurs a regret of*

$$\text{Reg}(T) = \tilde{O}\left(\sqrt{MK\tilde{d}T}\right)$$

*and the amount of real numbers communicated can be bounded as $O(d^\alpha \log(d^\alpha)\sqrt{M^3 T})$.*

### E.2 ROBUSTNESS, PRIVACY, AND BEYOND: DETAILS OF SECTION 6.2

We here provide some additional discussions on incorporating appendages in FL studies to provide robustness and privacy guarantees for FedIGW among some other directions, e.g., fairness guarantees (Mohri et al., 2019; Du et al., 2021), client selections (Balakrishnan et al., 2022; Fraboni et al.,

2021), and practical communication designs (Chen et al., 2021; Wei & Shen, 2022; Zheng et al., 2020). Following the unified principle that "**FCB = FL + CB**", we can develop the corresponding versions of FedIGW and the associated theoretical analyses following the comprehensive example of involving personalized learning in Section 6.1.

The key is that as long as one FL protocol can provide an estimated function $\widehat{f}$ (which is used in IGW interactions), it can be adopted in FedIGW; thus the desirable properties of the selected FL protocol are naturally inherited to FedIGW. For example, Yin et al. (2018); Pillutla et al. (2022); Fu et al. (2019); Li et al. (2021); Zhu et al. (2023) studied how to handle malicious agents, who can deviate arbitrarily from the FL protocol and tamper with their own updates, during learning. The commonly adopted approach is to invoke certain robust estimators (e.g., median and trimmed mean). Under suitable assumptions, existing approaches have shown that as long as the proportion of malicious agents does not exceed a threshold (typically, $1/2$), the estimators calculated by federation can still converge within certain amounts of error due to the malicious agents. A recent work (Zhu et al., 2023) provides a summary of convergence rates with different robust estimators, which can be leveraged to establish theoretical understandings of FedIGW with robustness.

On the privacy side, many mechanisms have also been studied in FL (Wei et al., 2020; Yin et al., 2021; Liu et al., 2022), to guarantee differential privacy (DP), where the most common approach is to insert noises of suitable scales. Convergence rates have also been established under suitable assumptions, e.g., in Wei et al. (2020); Girgis et al. (2021); Wei et al. (2021). With those analyses, the theoretical behavior of FedIGW with DP can also be similarly established as Corollaries D.7 and E.1.

# F  ALGORITHM SKETCHES AND CONVERGENCE ANALYSES OF FL DESIGNS

## F.1  FEDAVG

The FedAvg algorithm (McMahan et al., 2017) is one of the most standard and well-adopted FL protocol. Following it, agents perform local stochastic gradient descents (SGD) with their local objective functions for certain steps and then communicate the updated local models to the server; the server aggregates local models to a global one via a weighted average, which is then communicated to the agents to perform further local SGDs.

Many theoretical analyses have been provided for FedAvg (e.g., Li et al. (2020b)). We adopt the one from Karimireddy et al. (2020) in the following.

**Lemma F.1** (Theorem V in Karimireddy et al. (2020) without client sampling). *For any dataset $\mathcal{S}$, if*

- *$\widehat{\mathcal{L}}_m(f_\omega; \mathcal{S}_m)$ is $\mu_\omega$-strongly convex w.r.t. $\omega$ (see Definition F.2) for all $m \in [M]$;*

- *$\widehat{\mathcal{L}}_m(f_\omega; \mathcal{S}_m)$ is $\beta_\omega$-smooth w.r.t. $\omega$ (see Definition F.3) for all $m \in [M]$;*

- *the stochastic gradients are unbiased and have a $\sigma_b^2$-bounded variance (see Definition F.4);*

- *the gradients have $G_b$-bounded dissimilarity (see Definition F.5),*

*with FedAvg as the adopted FL protocol, the output $\widehat{\omega}$ satisfies that*

$$\mathbb{E}_\xi[\widehat{\mathcal{L}}(f_{\widehat{\omega}_\mathcal{S}}; \mathcal{S}) - \widehat{\mathcal{L}}(f_{\omega_\mathcal{S}^*}; \mathcal{S}) \mid \mathcal{S}] \leq \tilde{O}\left( \frac{\sigma_b^2}{\mu_\omega \rho \kappa M} + \frac{\beta_\omega G_b^2}{\mu_\omega^2 \rho^2} + \mu_\omega \|\omega^0 - \omega_\mathcal{S}^*\|_2^2 \exp\left( -\frac{\mu_\omega \rho}{16\beta_\omega} \right) \right)$$

*when $\rho \geq \frac{8\beta_\omega}{\mu_\omega}$, where $\rho$ denotes the round of communications (i.e., number of global aggregations), $\kappa$ is the number of local updates (i.e., SGD) between each communication, and $\omega^0$ is the initialization. Note that the last term which decays exponentially w.r.t. $\rho$ is omitted in Corollary D.6 and the following derivations for simplicity.*

A few definitions used above are made precise in the following, which are inherited from Karimireddy et al. (2020) and presented here for completeness:

**Definition F.2** (Strongly Convex). *$\widehat{\mathcal{L}}_m(f_\omega; \mathcal{S})$ is $\mu_\omega$-strongly convex w.r.t. $\omega$ for $\mu_\omega > 0$ if*

$$\widehat{\mathcal{L}}_m(f_{\omega'}; \mathcal{S}) - \widehat{\mathcal{L}}_m(f_\omega; \mathcal{S}) \geq \left\langle \nabla_\omega \widehat{\mathcal{L}}_m(f_\omega; \mathcal{S}), \omega' - \omega \right\rangle + \frac{\mu_\omega}{2} \left\| \omega' - \omega \right\|_2^2, \quad \textit{for any } \omega \textit{ and } \omega'.$$

**Definition F.3** (Smooth). *$\widehat{\mathcal{L}}_m(f_\omega; \mathcal{S})$ is $\beta_\omega$-smooth w.r.t. $\omega$ for $\beta_\omega > 0$ if*

$$\widehat{\mathcal{L}}_m(f_{\omega'}; \mathcal{S}) - \widehat{\mathcal{L}}_m(f_\omega; \mathcal{S}) \leq \left\langle \nabla_\omega \widehat{\mathcal{L}}_m(f_\omega; \mathcal{S}), \omega' - \omega \right\rangle + \frac{\beta_\omega}{2} \left\| \omega' - \omega \right\|_2^2, \quad \textit{for any } \omega \textit{ and } \omega'.$$

**Definition F.4** (Stochastic Gradients with Bounded Variances). *The stochastic gradients have a $\sigma_b^2$-bounded variance if*

$$\frac{1}{n_m} \sum_{i \in [n_m]} \left\| \nabla_\omega \ell_m(f_\omega(x_m^i, a_m^i); r_m^i) - \nabla_\omega \widehat{\mathcal{L}}_m(f_\omega; \mathcal{S}_m) \right\|_2^2 \leq \sigma_b^2, \quad \textit{for any } \omega \textit{ and } m.$$

**Definition F.5** (Gradients with Bounded Dissimilarity). *The gradients have a $G_b$-bounded dissimilarity if*

$$\frac{1}{M} \sum_{m \in [M]} \left\| \nabla_\omega \widehat{\mathcal{L}}_m(f_\omega; \mathcal{S}_m) \right\|_2^2 \leq G_b^2, \quad \textit{for any } \omega.$$

## F.2 SCAFFOLD

The SCAFFOLD algorithm is proposed in Karimireddy et al. (2020), which enhances FedAvg via leveraging variance reduction to correct drifts in heterogenous agents' local updates. The following result is established in Karimireddy et al. (2020) to characterize the convergence of the SCAFFOLD protocol.

**Lemma F.6** (Theorem VII in Karimireddy et al. (2020) without client sampling). *For any dataset $\mathcal{S}$, if*

- *$\widehat{\mathcal{L}}_m(f_\omega; \mathcal{S}_m)$ is $\mu_\omega$-strongly convex w.r.t. $\omega$ (see Definition F.2) for all $m \in [M]$;*

- *$\widehat{\mathcal{L}}_m(f_\omega; \mathcal{S}_m)$ is $\beta_\omega$-smooth w.r.t. $\omega$ (see Definition F.3) for all $m \in [M]$;*

- *the stochastic gradients are unbiased and have a $\sigma_b^2$-bounded variance (see Definition F.4),*

*with SCAFFOLD as the adopted FL protocol, the output $\widehat{\omega}$ satisfies that*

$$\mathbb{E}_\xi[\widehat{\mathcal{L}}(f_{\widehat{\omega}_\mathcal{S}}; \mathcal{S}) - \widehat{\mathcal{L}}(f_{\omega_\mathcal{S}^*}; \mathcal{S}) \mid \mathcal{S}] \leq \tilde{O}\left( \frac{\sigma_b^2}{\mu_\omega \rho \kappa M} + \mu_\omega \tilde{D}^2 \exp\left( -\min\left\{ \frac{\rho}{30}, \frac{\mu_\omega \rho}{162 \beta_\omega} \right\} \right) \right)$$

*when $\rho \geq \max\{\frac{162\beta_\omega}{\mu_\omega}, 30\}$, where $\rho$ denotes the round of communications (i.e., number of global aggregations), $\kappa$ is the number of local updates (i.e., SGD) between each communication, $\tilde{D}^2$ is a distant measure w.r.t. the initialization defined in Karimireddy et al. (2020). Note that the last term which decays exponentially w.r.t. $\rho$ is omitted in Corollary D.6 and the following derivations for simplicity.*

## F.3 LSGD-PFL

The LSGD-PFL protocol is summarized in Hanzely et al. (2021), which is a general design for personalized federated learning problems. It largely follows FedAvg (McMahan et al., 2017), while only the globally shared parameters are communicated and aggregated. The following lemma is provided in Hanzely et al. (2021) to characterize the convergence of LSGD-PFL.

**Lemma F.7** (Theorem 1 in Hanzely et al. (2021)). *For any dataset $\mathcal{S}$, if*

- *$\widehat{\mathcal{L}}_m(f_{\omega_m}; \mathcal{S}_m)$ is $\mu_\omega$-strongly convex w.r.t. $\omega_m$ (see Definition F.2) for all $m \in [M]$;*

- *$\widehat{\mathcal{L}}_m(f_{\omega^\alpha, \omega_m^\beta}; \mathcal{S}_m)$ is $\beta_{\omega^\alpha}$-smooth w.r.t. $\omega^\alpha$ and $M\beta_{\omega^\beta}$-smooth w.r.t. $\omega_m^\beta$ (see Definition F.3) for all $m \in [M]$;*

- *the stochastic gradients w.r.t. $\omega^\alpha$ is unbiased and have a $\sigma_b^2$-bounded variance (see Definition F.4);*

- *the stochastic gradients w.r.t. $\{\omega_m^\beta : m \in [M]\}$ is unbiased and have a $\sigma_b^2$-bounded variance (see Definition F.4);*

- *the gradients w.r.t. $\omega$ have $G_b$ bounded dissimilarity (see Definition F.5),*

*with LSGD-PFL as the adopted FL protocol, the output $\widehat{\omega}$ has $\varepsilon_{opt}(\mathcal{F}_{[M]}; n_{[M]}) \leq \varepsilon'$ after*

$$\tilde{O}\left( \frac{\max\{\beta_{\omega^\beta}\kappa^{-1}, \beta_{\omega^\alpha}\}}{\mu_\omega} + \frac{\sigma_b^2}{\mu_\omega \kappa M \varepsilon'} + \frac{1}{\mu_\omega}\sqrt{\frac{\beta_{\omega^\alpha}(G^2 + \sigma^2)}{\varepsilon'}} \right)$$

*rounds of communications, where $\kappa$ is the number of local updates.*

## G  EXPERIMENT DETAILS AND ADDITIONAL RESULTS

This section first provides a comprehensive description of the experimental settings and procedures. **The codes and detailed instructions have been uploaded in the supplementary materials so as to execute the experiments and reproduce the results.**

Additional results are also provided to deepen the understanding of the impact of adopting different FL protocols in FedIGW and the effect of involving different numbers of agents. Moreover, a performance comparison between FedIGW and the state-of-the-art FCB design, FN-UCB (Dai et al., 2023), is reported. These results reveal that our proposed FedIGW not only outperforms the single-agent designs but also supersedes the strong FCB baselines.

### G.1  EXPERIMENT SETTINGS

In the following, we report a comprehensive description of the experimental details adopted in the simulations.

**Datasets.** Our experiments employ two distinct real-world multi-label classification datasets, Bibtex (Katakis et al., 2008) and Delicious (Tsoumakas et al., 2008), which are also used in other practical CB investigations such as Cortes (2018). The aim of CB is considered to be recommending one of the correct labels at any given time. Especially, in the experiments, at each time step, a context is randomly sampled from the dataset while the true labels are concealed from the agents. The agents then determine which label to select (i.e., pull one arm) with their CB algorithms; thus the number of arms is the number of possible labels in each dataset. Upon pulling one arm, a reward of 1 is granted if the pulled arm corresponds to one of the true labels, while a reward of 0 is granted otherwise. Details of each task are listed in Table 3, from which we can observe that these scenarios are challenging given their high-dimensional contexts and large numbers of arms.

Table 3: The context dimension and number of arms in Bibtex and Delicious

| Task | Context dimension | Number of arms |
|---|---|---|
| Bibtex | 1835 | 159 |
| Delicious | 500 | 983 |

**Environments.** In the experiments, the environments sample contexts for all agents from the same set of datasets described above. For simplicity, the system is also designed as a synchronous one, i.e., $t_m(t) = t, \forall m \in [M]$.

**FedIGW.** As described in Section 5, for both tasks, two-layer multi-layer perceptrons (MLPs) with a hidden layer having a constant 256 width are used to approximate the reward functions in FedIGW. Moreover, multiple standard FL protocols including FedAvg (McMahan et al., 2017), SCAFFOLD (Karimireddy et al., 2020) and FedProx (Li et al., 2020a) are adopted as the FL component in FedIGW. During each FL process, the local batch size, the number of communications, and the local learning rate are specified in Table 4. Moreover, the epoch length is designed to be growing exponentially as in Corollaries 4.2, D.8 and E.2, i.e., $\tau^l = 2^l$, while culminating at an upper limit of 4096 to maintain timely updates.

Additionally, for practical conveniences, instead of selecting a theoretically-sound but sophisticated choice of $\gamma$ as in Theorem 4.1, we set it as a constant hyper-parameter and perform some preliminary manual selections with the final adopted values reported in Table 4. We believe this approach is more practically appealing as it does not need to scale $\gamma$ consistently; a similar choice of using constant $\gamma$'s is also adopted in Agarwal et al. (2023).

Table 4: Hyperparameter choices for FedIGW in Bibtex and Delicious

| Task | Learning Rate | Batch Size | Communications | Parameter $\gamma$ |
|---|---|---|---|---|
| Bibtex | 0.1 | 64 | 100 | 7000 |
| Delicious | 0.2 | 64 | 100 | 7000 |

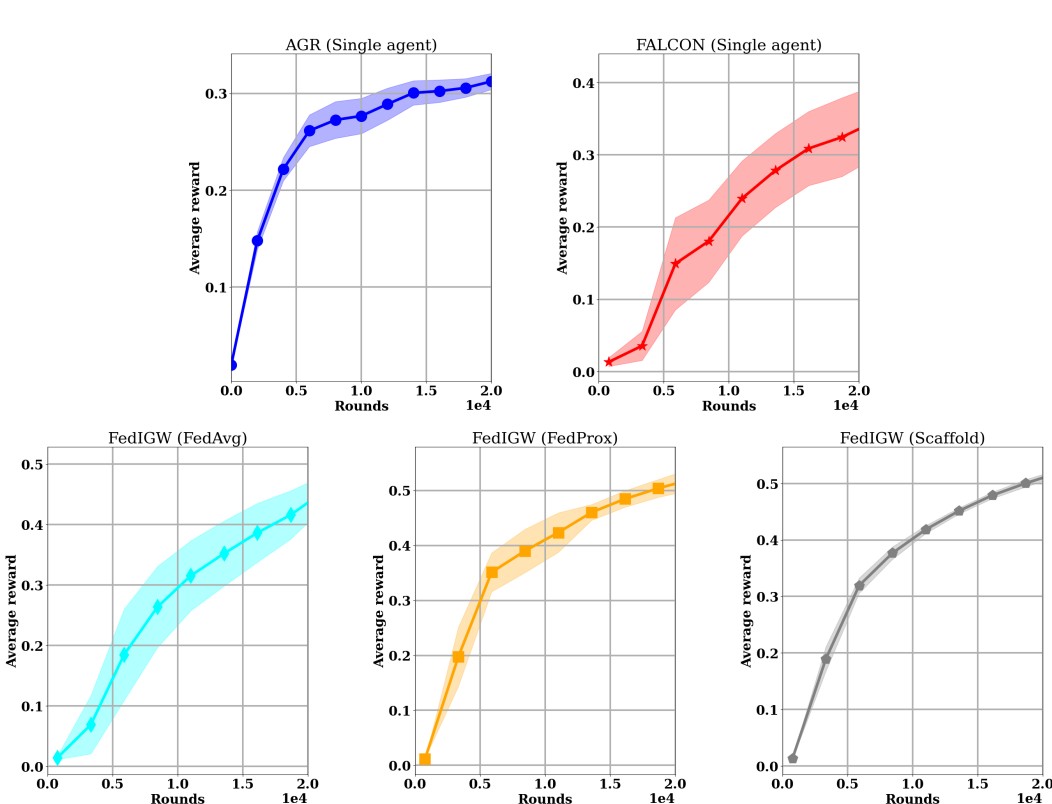

Figure 4: Averaged rewards and error bars on Bibtex from two single-agent baselines and FedIGW using FedAvg, FedProx, and SCAFFOLD as its FL protocol. The continuous curves represent the empirical average values, and the shadowed areas are the standard deviations.

**Single-agent baseline: AGR.** The adaptive greedy (AGR) algorithm (Chakrabarti et al., 2008) is selected as one of the single-agent baselines due to its strong empirical performance on Bibtex and Delicious reported in Cortes (2018). The algorithmic details can be found in Cortes (2018), and we also leveraged the code provided in Cortes (2018) to build this baseline.

**Single-agent baseline: FALCON.** The other single-agent baseline, FALCON, is proposed in Simchi-Levi & Xu (2022), which is essentially the single-agent version of FedIGW. We still adopt the same algorithmic configurations as FedIGW (i.e., epoch length, parameter $\gamma$, local batch size, and local learning rate) except that the MLP is optimized locally instead of in a federation, i.e., there are no communications.

**Performance evaluation.** All the reported results except Fig. 9 are averaged from 10 independent runs, whose error bars are further provided. In Fig. 9, the unpractical long running time of the baseline algorithm prohibits us from performing repeated experiments as discussed in the following description of computing resources.

**Computing Resources.** The computational requirement is relatively low for testing the two single-agent baselines (AGR and FALCON) and our proposed FedIGW. Specifically, we use a dual Nvidia-

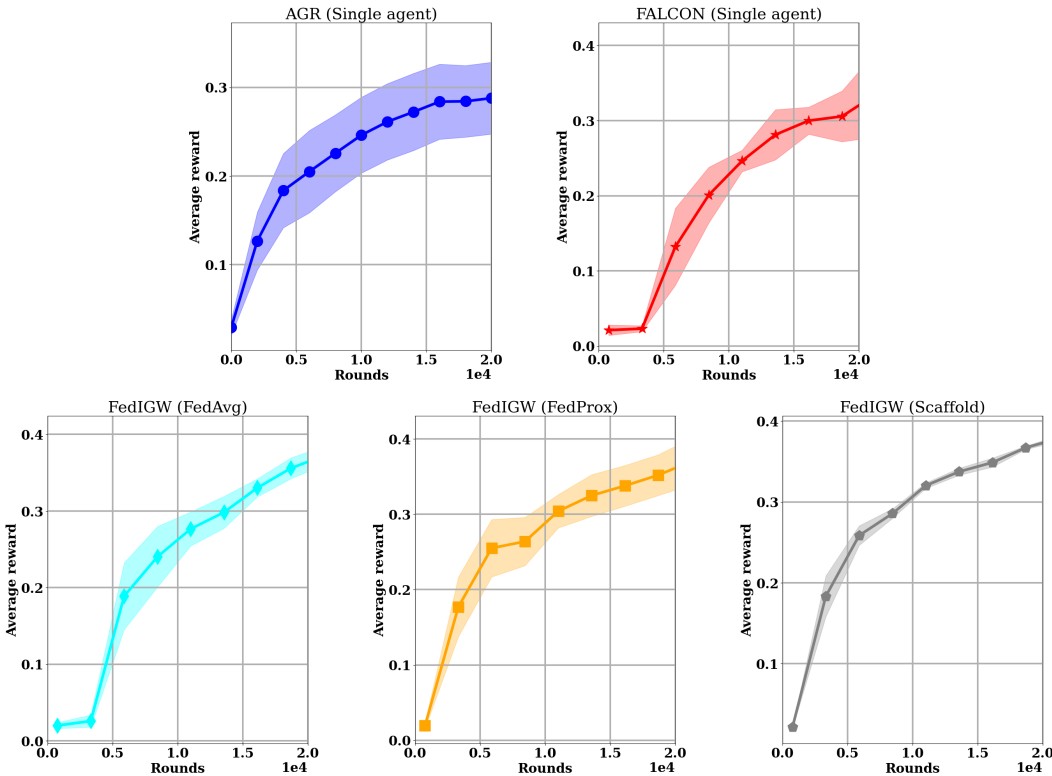

Figure 5: Averaged rewards and error bars on Delicious from two single-agent baselines and FedIGW using FedAvg, FedProx, and SCAFFOLD as its FL protocol. The continuous curves represent the empirical average values, and the shadowed areas are the standard deviations.

RTX 3090 workstation with an overall 20 GB RAM, which is more than needed as only 2 GB RAM is occupied during the experiments of AGR, FALCON, and FedIGW using the aforementioned MLP having a hidden layer with width 256. However, as illustrated later in Section G.4, when testing FN-UCB (Dai et al., 2023), the overall 20 GB RAM is not sufficient for running it when the MLP hidden layer has a width larger than 10; thus, we down-scaled the width to be 5 for smooth testing.

## G.2 ADDITIONAL RESULTS: VARYING FL CHOICES

We here provide additional details of Fig. 1 in Figs. 4 and 5, especially with error bars, and numerically verify the flexibility of FedIGW with different FL protocols, including FedAvg (McMahan et al., 2017), SCAFFOLD (Karimireddy et al., 2020) and FedProx (Li et al., 2020a). As observed in Section 5, using the further optimized SCAFFOLD and FedProx provides better performances compared with using the basic FedAvg, which credits to that FedIGW can seamlessly leverage algorithmic advances in FL protocols. Moreover, it can be observed that the performance obtained by using SCAFFOLD as the FL choice in FedIGW is also particularly stable, which demonstrates its superiority in the application of sequential decision-making.

## G.3 ADDITIONAL RESULTS: VARYING NUMBERS OF AGENTS

Fig. 6 further reports the averaged rewards of FedIGW with $M = 10, 20, 30, 50$ involved agents and their associated error bars in the Bibtex dataset, while Fig. 7 reports the same set of results in the Delicious dataset. Moreover, Fig. 8 compares the averaged rewards of FedIGW with varying numbers of agents. In these results, the FL protocol in FedIGW is selected to be FedAvg.

From Figs. 6 and 7, we can observe that FedIGW is capable of collecting more rewards than the two single-agent baselines, which demonstrates its efficiency. Moreover, Fig. 8 elucidates that the final performance of FedIGW is positively correlated with an increasing number of clients, which verifies the benefits of learning in a larger federation. Furthermore, the variance (i.e., error bar) in

Figs. 6 and 7 begins at a relatively small value and subsequently expands due to the initially intense explorations. Eventually, when the algorithm gradually approaches convergence, the variance begins to reduce, reflecting the stabilization of the learning process.

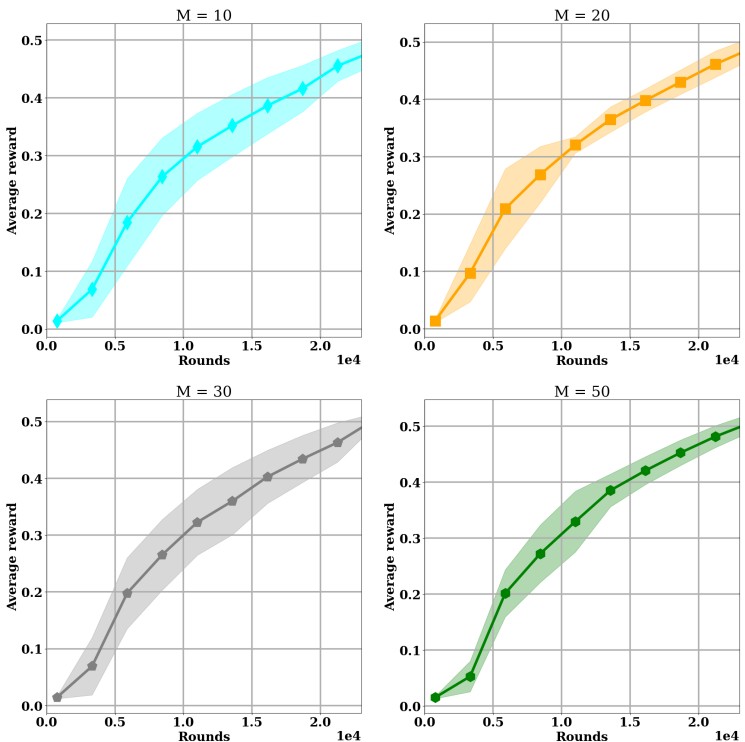

Figure 6: Averaged rewards and error bars on Bibtex from FedIGW (using FedAvg) with varying numbers of involved agents, i.e., $M = 10, 20, 30, 50$. The continuous curves represent the empirical average values, and the shadowed areas are the standard deviations.

### G.4 ADDITIONAL RESULTS: COMPARISON WITH FEDERATED NEURAL BANDITS

To further verify the performance of FedIGW, additional comparisons are conducted with a state-of-the-art FCB baseline, specifically, the federated neural-upper confidence bound (FN-UCB) design proposed in Dai et al. (2023). FN-UCB is capable of leveraging neural networks to approximate rewards and Dai et al. (2023) has reported superior performance compared to many other designs, which makes it a strong FCB baseline. When conducting the experiments, we first notice that FN-UCB necessitates multiple matrix inversions over the entire set of neural network parameters and such operations lead to substantial memory consumption when handling the high-dimensional context and numerous arms in both Bibtex and Delicious (which evidence the difficulty of these two employed datasets). To accommodate our already powerful computing resources (dual Nvidia-RTX 3090 and 20 GB RAM), we had to use a small-size MLP in both FALCON and FN-UCB for smooth testing and fair comparison. Especially, the width of the MLP hidden layer is down-scaled from the originally adopted 256 (in Fig. 1) to only 5, as our 20 GB RAM cannot support FN-UCB using an MLP with its hidden layer wider than 10. Also, due to the inefficiency of FN-UCB in our testing scenario, the error bars are omitted in Fig. 9. We believe this experimental observation demonstrates the computational efficiency of FedIGW over FN-UCB.

Moreover, the statistical performance of FedIGW and FN-UCB is presented in Fig. 9. As evident from the reported results, the substantial reduction in the neural network size has affected the performance of FedIGW compared with Figs. 6 and 7. In particular, FedIGW achieves only about 30% of the previously reported performance on Bibtex and 50% on Delicious. Nevertheless, despite this diminished performance, our proposed FedIGW still outperforms FN-UCB significantly, surpassing it by nearly 70% on both tasks as shown in Fig. 9. These comparisons further validate the advantages of FedIGW, demonstrating its relatively easy implementation and superior performance compared to existing FCB designs.

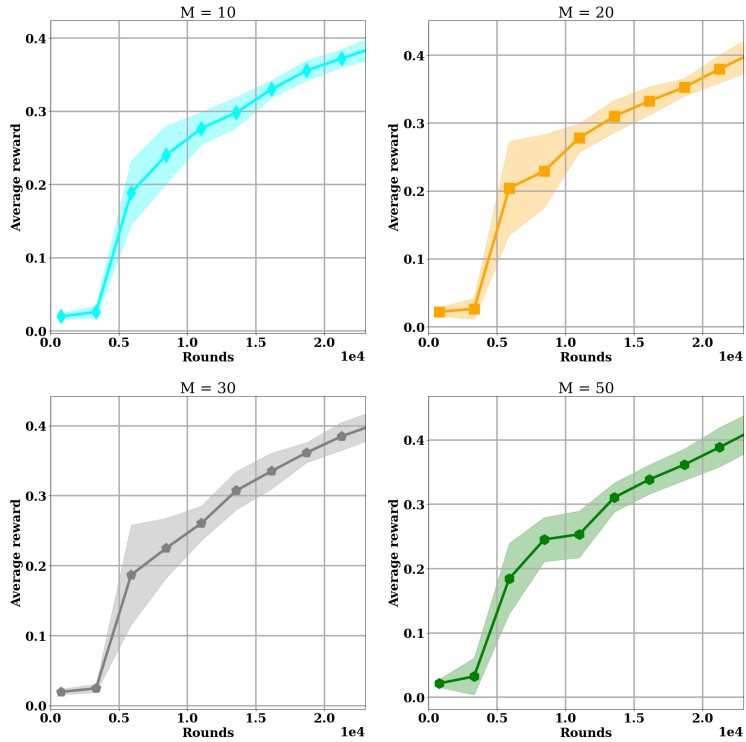

Figure 7: Averaged rewards and error bars on Delicious from FedIGW (using FedAvg) with varying numbers of involved agents, i.e., $M = 10, 20, 30, 50$. The continuous curves represent the empirical average values, and the shadowed areas are the standard deviations.

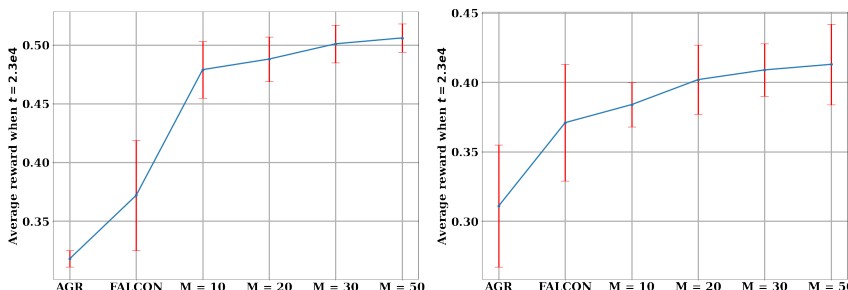

Figure 8: Averaged rewards and error bars on Bibtex (left) and Delicious (right) at time step $2.3 \times 10^4$ from two single-agent baselines and FedIGW (using FedAvg) with varying numbers of involved agents, i.e., $M = 10, 20, 30, 50$.

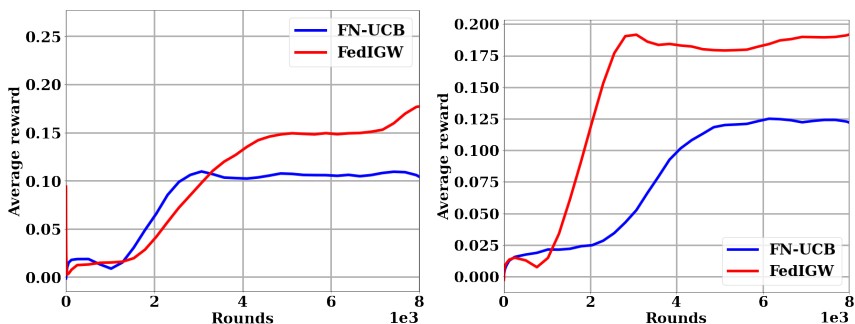

Figure 9: Comparisons between FedIGW and FN-UCB on Bibtex (left) and Delicious (right) with 10 involved agents.

