# OpenReview forum: "Harnessing the Power of Federated Learning in Federated Contextual Bandits"
_ICLR.cc/2024/Conference — ICLR 2024 Conference Withdrawn Submission_

### Official Review · Reviewer_Non5 · 2023-10-29

**Soundness:** 3 good
**Presentation:** 3 good
**Contribution:** 2 fair
**Rating:** 3
**Confidence:** 4

**Summary:**

This paper aims to leverage the results of FL in the supervised setting to design a unified framework for online federated bandits. To this end, it essentially builds upon the previous work of Simchi-Levi & Xu (2022) and generalizes it to the multi-agent setting.

**Strengths:**

It seems to be nice to have a unified view of FL bandits.

**Weaknesses:**

1. This framework seems to be a straightforward extension of single-agent work in Simchi-Levi & Xu (2022)
2. It also only handles the case where the contexts are stochastic, rather than adversary in previous works.

**Questions:**

1. What are the new challenges in the analysis? It seems to me that it is simply a combination of some simple algebras so as to generalize to the multi-agent case.
2. How to handle the adversary context?
3. One may need to be careful when making claims that the proposed framework can handle DP easily. In the bandit case, the DP definition differs from the supervised learning setting in terms of the neighboring datasets. Without a careful definition of DP, it is not clear how the current framework can provide the right protection and performance analysis.

---

### Official Review · Reviewer_jNg7 · 2023-10-30

**Soundness:** 3 good
**Presentation:** 3 good
**Contribution:** 2 fair
**Rating:** 5
**Confidence:** 3

**Summary:**

This paper studies the federated contextual bandit (FCB) problem, which is a combination of federated learning (FL) and contextual bandits (CB). Specifically, each agent can share information together with other agents through a central server and benefit from the shared information in solving its local contextual bandit problem. The paper discusses the compatibility between the FL part and CB part in the FCB design and proposes a federated inverse gap weighting (FedIGW) algorithm to solve the FCB problem, which is an extension of the inverse gap weighting (IGW) from conventional CB problems to FCB. Theoretical results on performance bounds are provided for the FedIGW, along with numerical results.

**Strengths:**

The presentation is clear. It is easy to follow the setting of FCB, the FedIGW algorithm, and the main theoretical and numerical results. The flow charts and highlights are also helpful.

**Weaknesses:**

1. Table 1 is a good summary of the current SOTA FCB works with their respective FL and CB components, but it is unclear how the convergence rates and regret compare to this work.
2. The improvement over the previous works in FCB seems marginal, it seems to be replacing the UCB or other no-regret CB algorithm with IGW in the local CB part with marginal benefits including "avoiding complicated data analysis in UCB". It will be better for the authors to further elaborate on the advantages (e.g., convergence rate, regret, compatibility) of FedIGW over FL + UCB designs.
3. I can not locate a definition, proposition, or theorem that discusses the compatibility/incompatibility of the FL and CB parts. This makes the reader unclear as to why not combine any SOTA FL and CB design to solve FCB and have to stick with FedIGW.
4. The problem is limited to a finite action set, which I think could be a limitation of IGW, and the potential issue of discretizing continuous action space is not discussed.
5. No confidence interval for the plots, this is a necessity. The authors should consider adding intervals with high transparency (e.g., low alpha values in Python) in the figures in the main article.

**Questions:**

Besides the weaknesses, I have the following questions

1. For a given system with a known number of agents, data heterogeneity bounds, and known communication cost, if FedIGW is not always the optimal option, is there a systematic approach to find the optimal or approximately optimal choice of FL and CB components?
2. Will an alternative design where each agent applies different CB algorithms locally benefit the system? If not, what is the main problem with the alternative approach?

---

### Official Review · Reviewer_YJux · 2023-11-02

**Soundness:** 3 good
**Presentation:** 3 good
**Contribution:** 2 fair
**Rating:** 5
**Confidence:** 3

**Summary:**

The paper proposes a flexible framework (FedIGW) for federated contextual bandits. Unlike most existing algorithms in FCB that require compressed local data to be aggregated at server, the proposed algorithm aggregates the local models (i.e., the estimated reward function of each client). This allows the method to decouple CB and FL components and adopt versatile FL protocols. The paper adopts IGW as CB component to estimate local reward function, and conduct the theoretical analysis of the proposed algorithm. The paper also conducts experiments to validate the proposed method with different FL protocols.

**Strengths:**

1. The paper provides theoretical analysis of the proposed algorithm. Notably, the analysis can simply leverage the analysis of any FL protocols as a plug-in.
2. The paper is well-written and easy to follow. The summery of related works in Table 1 is helpful.

**Weaknesses:**

1. One advantage of federated learning is that it allows the local computation to be decoupled from the global aggregation, i.e., the server may use various global aggregation rule to aggregate clients’ local updates, and clients can use different methods to update their local models. Different combinations of local computation and global aggregation result in different algorithms. I believe such flexibility also holds naturally for contextual bandits: if we let each client’s local model be client’s reward function, then the clients can use different methods to update their reward functions and the server can use different rules to aggregate the local reward functions. Since the main advantage of FedIGW is that it allows us to use versatile FL protocols, it is unclear to me why this idea of having decoupled components of CB and FL is novel in FCB.
2. In FedIGW, we can use arbitrary FL protocols but the CB component has to be IGW. I believe any other methods should also work, as long as it can estimate reward function. Are there any other reasons for using IGW? Since IGW and the corresponding theoretical analysis were proposed by prior works, the contribution of this paper is somewhat marginal.
3. Decoupling CB and FL in FCB has been proposed in Agarwal et al. (2023), and the authors argue this paper has additional contributions by having theoretical analysis. However, most of their theoretical analysis is adopted directly from IGW-type CB algorithms.
4. The experiments are not sufficient. The authors only evaluate the proposed algorithm (with different FL protocols) without comparing it with other FCB algorithms. More extensive comparison with other FCB algorithms, especially the one in Agarwal et al. (2023) which also decouples CB and FL, is necessary.

**Questions:**

1. How is the realizability assumption used in the analysis? How can the regret bound be impacted when the assumption is violated?
2. Can you explain why it is difficult to decouple FL and CB in FCB? I had thought this idea is very straightforward.
3. See other concerns in the weakness section.